# Plantain: Plan-Answer Interleaved Reasoning

Anthony Liang [* 1 2]   Jonathan Berant [1]   Adam Fisch [1]   Abhimanyu Goyal [1]   Kalpesh Krishna [† 1]
Jacob Eisenstein [† 1]

## Abstract

Reasoning models often spend significant time generating hidden reasoning before any visible response, which can waste user time when the model starts from a false premise that could have been corrected early. Human speakers, in contrast, use lightweight incremental check-ins to maintain common ground, motivating *interleaved reasoning* (IR), where a model alternates between internal thinking and visible intermediate responses. We instantiate this idea with PLANTAIN (Plan-Answer Interleaved Reasoning), a post-training recipe that teaches a model to externalize an explicit step-by-step plan before continuing its reasoning. This learned plan-first structure creates an interface for early feedback and intervention while preserving space for subsequent reasoning. Across challenging math, coding, text-to-SQL, and reading-comprehension benchmarks, PLANTAIN improves pass@1 by roughly 6% on average while reducing time-to-first-response by over 60% relative to think-then-answer baselines.

## 1. Introduction

Reasoning models ([Jaech et al., 2024](#); [Guo et al., 2025](#); [Yang et al., 2025](#)) typically take a "think-then-answer" approach, in which a monolithic, and often quite long, block of reasoning is generated before producing any user-facing output. This delay can result in substantial wasted time if the model misunderstands the user's intent or begins its reasoning from a flawed assumption. Because it offers no opportunity for user intervention, the "think-then-answer" paradigm is especially problematic in time-sensitive applications like voice assistants.

---
[*]Lead author. [†]Co-senior authors. [1]Google DeepMind, London, United Kingdom [2]Thomas Lord Department of Computer Science, University of Southern California, Los Angeles, CA, USA. Correspondence to: Anthony Liang <aliang80@usc.edu>.

*Proceedings of the 43rd International Conference on Machine Learning*, Seoul, South Korea. PMLR 306, 2026. Copyright 2026 by the author(s).

Human speakers avoid this problem by employing a range of strategies for *collaborative grounding*, which help to ensure mutual understanding throughout dialogue ([Clark & Schaefer, 1989](#); [Benotti & Blackburn, 2021](#); [Shaikh et al., 2023](#)). Motivated by this literature, we introduce *interleaved reasoning*, in which the model alternates between unobserved "thinking" and surfacing intermediate responses to the user. We define an intermediate response as a self-contained, usable piece of information that hints at the model's understanding of the user's intention and its plan for satisfying it. When this understanding or plan is incorrect, it should be possible for the user to act immediately without waiting for a future revision. For example, in response to a trip planning prompt, the model might first provide a high-level outline of a possible itinerary. Subsequent responses could then provide more granular, day-to-day activities, after making additional web searches and tool calls.

PLANTAIN is therefore not a prompting-only intervention: the model is post-trained to externalize a plan and then continue reasoning conditioned on that surfaced plan. PLANTAIN follows a three-stage recipe: (1) generating a synthetic dataset of interleaved reasoning traces using prompting, (2) supervised fine-tuning to distill the desired format into the base model, and (3) reinforcement learning (RL) post-training with verifiable rewards for improving downstream task performance. In this paper, we focus on a "plan-first" specialization of interleaved reasoning, shown in Figure 1, where the model's initial intermediate response is an explicit, *step-by-step* plan that verbalizes its intended solution path.

While prior work has shown that grounding model generation in a plan improves performance ([Yao et al., 2023](#)), our interleaved approach uniquely allows for early intervention to keep the solution on track. To this end, we design two inference-time strategies, **Best-of-N-Plans** and **Rewind-and-Repeat**, that use an LLM-as-a-judge to simulate user feedback on the initial plan(s). In the Best-of-N-Plans approach, the model generates multiple plans and the user selects the best one for the subsequent generation. In Rewind-and-Repeat ([Kim et al., 2024](#)), the model proposes one plan at a time, which the user can accept or reject; if a plan is rejected, the process is "rewound", and the model is prompted to generate a new, different plan. We evaluate these methods on a diverse set of benchmarks, including

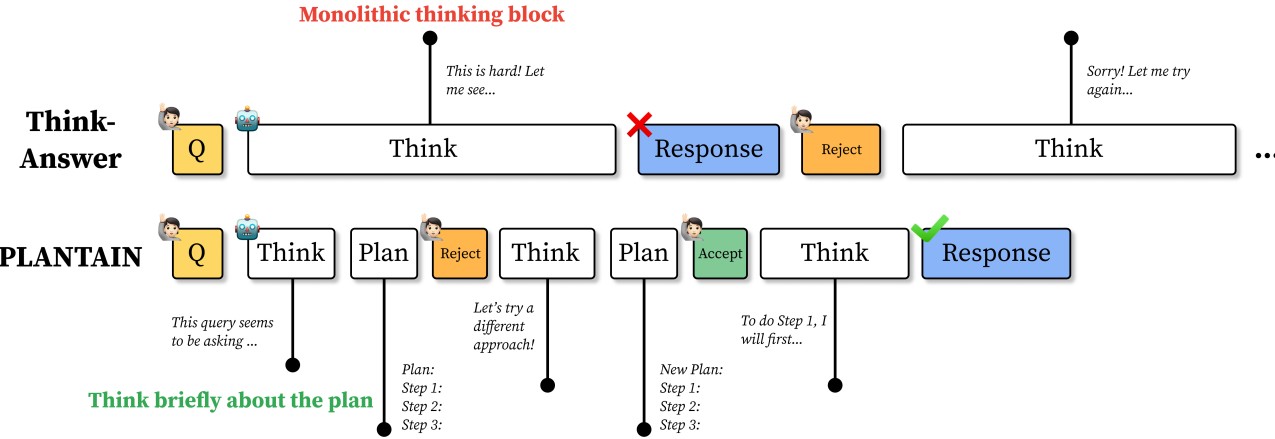

*Figure 1.* **PLANTAIN Rewind-and-Repeat** After post-training an interleaved reasoning model, we apply an iterative rejection-sampling strategy for inference-time plan guidance. PLANTAIN first produces a *plan* as its initial intermediate response. This plan is evaluated by a human judge or an LLM autorater. If accepted, the model proceeds to generate the final answer. If rejected, the process is *rewound* such that the rejected plan is appended to a history of failures, and the model is re-prompted to produce a new, distinct plan. Crucially, the response generated by PLANTAIN is typically of higher quality than the response produced by a standard Think–Answer model. Additionally, subsequent thinking blocks after a rejection plan are short because the model only needs to adjust the plan rather than regenerate full reasoning. This leads to early pruning of suboptimal reasoning paths and closer alignment with true user intent.

math reasoning, coding, text-to-SQL structured generation, and long-context question answering. Our proposed strategies improve alignment and robustness without increasing total token cost, since only short plan prefixes are resampled rather than full generations. This allows the model (or user) to prune flawed reasoning paths early, yielding higher pass@1 and lower time-to-first-response (TTFR) compared to standard Best-of-N over complete responses (Shen et al., 2004).

Our core contributions are as follows:

1. We introduce PLANTAIN, a framework for "plan-first" interleaved reasoning, along with two inference-time strategies that allow for early user intervention and feedback.
2. We demonstrate that PLANTAIN reduces time-to-first-response by **60%**, without compromising task performance.
3. We demonstrate that PLANTAIN—when trained exclusively on coding data—generalizes to diverse reasoning tasks, achieving an average **+6%** improvement in pass@1 across all benchmarks.

Our experiments study this plan-first interface in a controlled setting with an LLM autorater standing in for user feedback. We therefore treat TTFR as a measured proxy for the time until useful visible output, rather than as a direct measurement of perceived latency. The broader goal is to test whether post-training can create an early intervention point that improves the latency–accuracy tradeoff; evaluating PLANTAIN with human users and richer agentic tasks remains an important direction for future work.

**Conflict of Interest Disclosure.** Some authors are employed by Google DeepMind. This work evaluates open-weight models not developed by Google DeepMind, and the authors declare no financial conflicts of interest beyond their listed affiliations.

## 2. Interleaved Reasoning

PLANTAIN is a post-training framework to elicit interleaved reasoning behavior in reasoning models. Unlike prior approaches that impose length restrictions or introduce token budgets (Shen et al., 2025; Aggarwal & Welleck, 2025; Han et al., 2024), PLANTAIN encourages the model to produce an explicit *plan* as the first intermediate response. This behavior is controlled through the system instruction (SI), which is modified to prompt the model to "plan first" (shown in Section A.2), while the SFT and RL objectives reinforce the resulting interleaved reasoning style without explicitly constraining when to plan. Our framework is a simple three-step recipe: (1) generation of synthetic interleaved reasoning traces, (2) supervised fine-tuning to distill the interleaved format, and (3) RL post-training using verifiable rewards. Having obtained models capable of interleaved reasoning, we then introduce two inference-time strategies that leverage this output structure to improve the final response generation. The complete pseudocode for our training and inference procedures is provided in Section A.1.

**Interleaved CoT.** A standard language model is typically trained to produce a final answer $A$ conditioned on a prompt $P$, often preceded by a monolithic block of reasoning, $T$, resulting in a sequence of the form, $P \rightarrow T \rightarrow A$. Rather than

---

**Algorithm 1** Post-training to elicit interleaved reasoning

---

**Input:** Base model $\pi_{\theta_0}$; prompts $\mathcal{X}$; template $\text{SI}_{\text{plan-first}}$; synthetic generator $\Pi^\star$; reward weights $\boldsymbol{\alpha}$; number of RL updates $K$.

**Output:** Interleaved model $\pi_\theta$

1: // Construct synthetic interleaved trace dataset.
2: $\mathcal{D}_{\text{interleave}} \leftarrow$
   $\{(x', \tau) \mid x' = \text{SI}_{\text{plan-first}}(x),\ \tau \sim \Pi^\star(\cdot \mid x'),\ x \in \mathcal{X}\}$
3: // SFT on the synthetic interleaved trace dataset.
4: $\textsc{SupervisedFineTune}(\pi_{\theta_0} \rightarrow \pi_\theta;\ \mathcal{D}_{\text{interleave}})$
5: // RL post-training to obtain $\textsc{Plantain}$ model.
6: **for** $k = 1$ to $K$ **do**
7: $\quad$ Sample $(x', \cdot) \sim \mathcal{D}_{\text{interleave}}$; rollout $y \sim \pi_\theta(\cdot \mid x')$
8: $\quad r_{\text{fmt}} \leftarrow \textsc{FormatOK}(y)$
9: $\quad r_{\text{corr}} \leftarrow \textsc{UnitTestPassRate}(x', y)$
10: $\quad r_{\text{help}} \leftarrow \textsc{LLMJudge}(x', y)$
11: $\quad r_{\text{unit}} \leftarrow \mathbf{1}\{\text{valid unit-test block in } y\}$
12: $\quad r_{\text{aux}} \leftarrow \alpha_{\text{corr}} r_{\text{corr}} + \alpha_{\text{help}} r_{\text{help}} + \alpha_{\text{unit}} r_{\text{unit}}$
13: $\quad r \leftarrow \alpha_{\text{fmt}} r_{\text{fmt}} + r_{\text{fmt}} r_{\text{aux}}$
14: $\quad \textsc{PerformRLUpdate}(\theta; x', y, r)$
15: **return** $\pi_\theta$

---

outputting the final answer after a lengthy thought block, interleaved CoT alternates between shorter thoughts and intermediate responses surfaced to the user. Formally, an interleaved trace is $\tau = (t_1, a_1, t_2, a_2, \ldots, t_n, a_n)$, where $t_i$ represents internal thought and $a_i$ is a user-visible intermediate response. In the plan-first specialization studied here, the first surfaced response $a_1$ is an explicit plan $z$, and the remainder of the response is generated conditionally as $y_{\text{rem}} \sim \pi_\theta(\cdot \mid P, z)$.

**Generating Synthetic Interleaved Traces.** Off-the-shelf reasoning models exhibit a strong bias towards the "think-answer" paradigm, an artifact of their training that results in lengthy reasoning sequences even for simple prompts (Chen et al., 2024). Following (Kang et al., 2025), we iteratively prompt a larger model to produce variable length CoT traces. We take a subset of prompts from BIGCODEBENCH (Zhuo et al., 2024) and create a natural response decomposition, by modifying the prompt (Sections A.7 and A.8) to ask for a solution outline and unit tests in addition to the code implementation. Concretely, each interleaved CoT trace follows the structure *thought → solution plan → thought → code → thought → unit tests*, encouraging the model to emit useful early outputs for the user. We provide examples of the generated interleaved traces in Section A.8. To improve generalization, we further construct (i) *concatenated-prompt traces*, where multiple independent prompts are concatenated into a composite input, and the model is asked to solve each sequentially within a single reasoning trace, and (ii) *multi-solution traces*, where the model is prompted to generate several distinct candidate solutions for the same problem. For example, a concatenated-prompt trace may include three standalone coding tasks, such as reversing a string, counting unique elements, and checking for palindromes, encouraging the model to transition between problems within one

reasoning context. A multi-solution trace could prompt the model to produce multiple implementations for a single task (e.g., recursive and iterative factorial). Together, these examples form our synthetic interleaved reasoning dataset, denoted as $\mathcal{D}_{\text{interleave}} = \{(x'_j, \tau_j)\}_{j=1}^{N}$, where $x'_j = \text{SI}_{\text{plan-first}}(x_j)$. This construction diversifies the amount of multi-step interleaving in our training data and prevents the model from overfitting to a fixed output template.

We fine-tune the base model on $\mathcal{D}_{\text{interleave}}$ by minimizing the negative log-likelihood,

$$\mathcal{L}_{\text{SFT}}(\theta) = - \sum_{(x', \tau) \in \mathcal{D}_{\text{interleave}}} \log \pi_\theta(\tau \mid x').$$

This process effectively "distills" the desired interactive behavior from our synthetic data into the model, shifting its default response style from monolithic to interleaved. We then post-train the supervised fine-tuned model using Proximal Policy Optimization (PPO) (Schulman et al., 2017). PPO utilizes a value network to approximate the state-value function and Generalized Advantage Estimation to compute the advantage function. (Xie et al. (2025) found PPO to be more stable during training compared to GRPO (Guo et al., 2025) because of its extra critic model.) The policy model $\pi_\theta$ generates rollouts that maximize an expected reward $\mathbb{E}[r(x, y)]$.

**Reward function.** For our modified coding prompts, we define a composite reward comprised of four components: a **format** reward that checks whether the response correctly interleaves multiple intermediate answers, a **correctness** reward based on the pass rate of the generated code against golden unit tests, a **helpfulness** reward produced by an LLM-as-a-judge autorater evaluating the quality of the outline, and a **unit-test** reward indicating whether a valid unit-test block was produced. The overall reward is computed as: $r_{\text{interleave}}(x, y) = \alpha_{\text{fmt}} r_{\text{format}}(y) + r_{\text{format}}(y)\big(\alpha_{\text{corr}} r_{\text{corr}}(x, y) + \alpha_{\text{help}} r_{\text{help}}(x, y) + \alpha_{\text{unit}} r_{\text{unit}}(y)\big)$, where

$$r_{\text{format}}(y) = \mathbf{1}\{\text{response contains all of the required}$$
$$\text{sections in the correct order}\},$$
$$r_{\text{corr}}(x, y) = \frac{\#\text{ tests passed by } y}{\#\text{ total tests for } x},$$
$$r_{\text{help}}(x, y) = \text{LLM-Judge}(x, y) \in [0, 1],$$
$$r_{\text{unit}}(y) = \mathbf{1}\{\text{valid unit-test block detected}\}.$$

By including $r_{\text{format}}(y)$ as a multiplicative gate, we ensure that downstream rewards are applied only when the interleaved format is satisfied, while still giving a direct reward for producing the required plan-first structure. In preliminary experiments, we find that naively modifying the System Instruction (SI) does not induce the desired interleaved behavior. Models prompted in this way often collapse the plan generation into the reasoning trace and revert to the standard monolithic *think-then-answer* format. This structure

is essential for eliciting consistent plan-first reasoning and enabling our subsequent inference-time strategies.

## 3. Inference-Time Scaling

Given a trained interleaved reasoning model, we can apply inference-time strategies to better align model outputs with user intent, pruning erroneous reasoning paths early in the generation process. These strategies operate at inference without modifying the underlying model weights, leveraging the model's ability to produce an explicit *plan* as the first intermediate response.

While we envision these approaches having utility in inter-active settings with human users, in this paper we simulate the human with an autorater that provides feedback on candidate plans (prompts shown in Section A.3). The judge receives as input the user's original prompt $p$ and a proposed plan $a_i$; because it has access to the user prompt and plan but not the model's internal reasoning trace, it evaluates plans solely from the perspective of an external user. Given $N$ candidate plans $\{a_i\}_{i=1}^N$, the judge is prompted to either (i) select the index of the best plan when $N > 1$, or (ii) output a binary decision $\{\texttt{accept}, \texttt{reject}\}$ when evaluating a single plan ($N = 1$). We use $R_{\text{select}}(p, \{a_i\}_{i=1}^N)$ to denote the selected index for Best-of-$N$ and $R_{\text{bin}}(p, a_i)$ to denote the binary accept/reject verdict used for R&R.

Given that our model can interleave between internal thinking and user-facing responses, we use the first intermediate output as a *plan*. We introduce two inference-time control methods based on this plan structure: (1) *Best-of-N-Plans*, which samples multiple candidate plans and uses the LLM autorater to select the plan that best addresses the user prompt, and (2) *Rewind-and-repeat*, which generates a new plan whenever the current one is rejected, continuing until an acceptable plan is found or a retry budget is exhausted. Together, these strategies enable adaptive, user-aligned reasoning at inference time without retraining. Although the model is initially trained on coding tasks, these inference-time mechanisms are domain-agnostic and extend naturally to other reasoning settings such as mathematics, text understanding, or planning. Pseudocode for the inference-time strategies is provided in Algorithms 2 and 3.

**Best-of-N Plans.** In this setting, the model generates a diverse set of $N$ candidate plans, $\{a_1, a_2, \ldots, a_N\}$, for a given user prompt $p$. To encourage diversity, we sample with a high temperature ($\tau > 1$), which increases the likelihood of exploring less frequent but potentially insightful reasoning paths. The generated plans are then evaluated by the LLM *autorater*, which serves as a proxy for human feedback. The autorater receives the user's prompt $p$ along with the set of candidate plans, but not the model's internal reasoning process, introducing *information asymmetry*

---

**Algorithm 2** Best-of-N-Plans

**Input:** Interleaved model $\pi_\theta$; prompt $p$; temperature $\tau$; number of plans $N$; AutoRater $R_{\text{select}}$ for selecting plans.
**Output:** Final response $y$.
1: // Convert $p$ to a "plan-first" prompt.
2: $p' \leftarrow \texttt{SI}_{\text{plan-first}}(p)$
3: // Sample and score $N$ plans.
4: Sample $a_i \sim \pi_\theta(a \mid p'; \tau)$ for $i = 1, \ldots, N$
5: // Select the best auto-rated plan.
6: $i^\star \leftarrow R_{\text{select}}(p, \{a_i\}_{i=1}^N); \quad a^\star \leftarrow a_{i^\star}$
7: // Complete rollout according to the plan, and answer.
8: $y \sim \pi_\theta(\cdot \mid p', a^\star)$
9: **return** $y$

---

**Algorithm 3** Rewind-and-Repeat (R&R)

**Input:** Interleaved model $\pi_\theta$; prompt $p$; number of retries $T$; AutoRater $R_{\text{bin}}$ for accepting or rejecting plans.
**Output:** Final response $y$.
1: // Initialize the prompt and rewind history.
2: $p' \leftarrow \texttt{SI}_{\text{plan-first}}(p); \quad \mathcal{H} \leftarrow \varnothing$
3: **for** $t = 1$ to $T$ **do**
4:     // Propose and rate a plan.
5:     $c_t \leftarrow \text{AUGMENT}(p', \mathcal{H}); \quad a_t \sim \pi_\theta(a \mid c_t)$
6:     $d_t \leftarrow R_{\text{bin}}(p, a_t)$
7:     **if** $d_t = \texttt{accept}$ **then**
8:         // Accept plan, complete rollout, and answer.
9:         **return** $y \sim \pi_\theta(\cdot \mid c_t, a_t)$
10:    **else**
11:       // Store the rejected plan in the rewind history $\mathcal{H}$.
12:       $\mathcal{H} \leftarrow \mathcal{H} \cup \{a_t\}$
13:    **end if**
14: **end for**
15: // Fallback to a past plan from $\mathcal{H}$ if all attempts are rejected.
16: $a_r \sim \text{Uniform}(\mathcal{H}); \quad$ **return** $y \sim \pi_\theta(\cdot \mid p', a_r)$

---

between the model and autorater. This setup mirrors real user evaluation, where one can judge whether a plan is well-structured and relevant to the prompt without access to the model's internal thoughts. The autorater selects the index $i^\star = R_{\text{select}}(p, \{a_i\}_{i=1}^N)$ of the plan that best addresses the prompt and is most likely to lead to a correct or complete final response, denoted as $a^\star = a_{i^\star}$. The model then continues generation conditioned on $a^\star$, producing the subsequent reasoning steps and final answer.

**Rewind-and-repeat.** Instead of generating multiple plans at once, this approach performs plan generation in an iterative feedback loop. The model first proposes an initial plan $a_1$ conditioned on the user prompt $p$. The LLM autorater then evaluates the plan and returns a binary verdict $d_1 = R_{\text{bin}}(p, a_1) \in \{\texttt{accept}, \texttt{reject}\}$. If accepted, the model proceeds to complete the response using $a_1$. If rejected, the plan is added to the failure set $\mathcal{H} = \mathcal{H} \cup \{a_1\}$, and the model is re-prompted with the original query and rejection history, written as $c_t = \text{AUGMENT}(p', \mathcal{H})$ at retry $t$. This conditioning encourages the model to generate a new plan that avoids previous failure modes. The process

repeats for up to $T$ iterations or until a plan is accepted. If all attempts are rejected, the model samples a fallback plan from $\mathcal{H}$ and continues generation from that plan.

# 4. Experimental Setup

Our experiments address two questions: **(Q1)** Does interleaved reasoning reduce time-to-first-response without compromising final task performance compared to standard think-answer decoding? **(Q2)** How effective are inference-time strategies at improving initial plan quality and pruning suboptimal reasoning paths before execution?

**Models and Baselines.** We use the Qwen3 reasoning model family (Yang et al., 2025) as our base architecture, and evaluate both the 4B and 8B parameter variants. We compare our post-trained interleaved reasoning models and inference-time strategies against several baselines and ablations. The first set of baselines is applied directly to the base model without any additional fine-tuning.

- **No Thinking:** The base model is prompted to generate the answer directly, without explicit CoT reasoning.
- **Think-Answer (TA):** The base model is prompted to perform explicit reasoning steps in a monolithic block before generating the final answer.
- **Rewind-and-Repeat (R&R) on Answer:** Rewind-and-repeat applied to the *final answer*. An LLM autorater judges the final response and then triggers a full restart from the original prompt if the answer is rejected. This inference strategy is equivalent to our proposed $R\&R$, but applied at the final response level.

We compare the baselines to our inference-time strategies applied on the RL-trained interleaved reasoning model.

- **Plan-Answer:** Direct inference of the interleaved thinking model. First generate a plan, and then the final answer.
- **Best-of-N Plan:** Use the RL fine-tuned model to generate $N$ plans in parallel, and select the best one using an LLM autorater. The final response is then conditioned on the selected Best-of-N plan.
- **Rewind-and-Repeat (R&R) Plan:** Iterative rejection-sampling approach that generates a single plan at a time, with an LLM autorater to decide whether to accept it and continue, or reject and "rewind" to generate a new plan.

All interleaved decoding runs use the plan-first system instruction in Section A.2, and all plan-level inference strategies use the LLM plan autorater prompt in Section A.3. Training hyperparameters and compute details are provided in Section A.6.

**Interleaved and Evaluation Datasets.** We train on a combination of coding and mathematical reasoning datasets (BigCodeBench, MBPP, and MATH500). To assess cross-domain generalization we evaluate across two additional domains: text-to-SQL and long-context question answering. BigCodeBench (Zhuo et al., 2024) contains 1.1K function-level Python coding tasks requiring multi-library reasoning and compositional code synthesis. MBPP (Austin et al., 2021) consists of 974 entry-level Python problems designed for evaluating basic programming competence. MATH500 (Lightman et al., 2023) includes 500 symbolic reasoning problems spanning algebra, geometry, and probability. BirdSQL (Li et al., 2023) and QuALITY (Pang et al., 2021) are used for out-of-domain evaluation: the former tests text-to-SQL translation grounded in real relational schemas, while the latter measures long-context reading comprehension with passages averaging 5K tokens. See Sections A.4 and A.5 for dataset sizes, splits, and full dataset descriptions. We generate interleaved traces on 50 BigCodeBench prompts modified to request a solution outline and unit tests in addition to the code solution. Iterative CoT prompting on Qwen3-32B is used to synthesize these traces. We additionally collect 50 traces by concatenating pairs of MBPP prompts and 25 traces that request multiple solutions for a single coding problem, resulting in 125 total interleaved reasoning traces used for SFT. The synthetic generation procedure and prompt examples are shown in Sections A.7 and A.8.

**Evaluation Metrics.** We evaluate model performance using four primary metrics: **task success rate (pass@1)**, **unit test pass rate**, **time-to-first-response (TTFR)**, and **tokens-per-problem (T/P)**. Pass@1 measures the percentage of prompts for which the model produces a fully correct solution on the first attempt and is reported across all benchmarks. For coding tasks, we additionally report the unit test pass rate, the proportion of ground-truth unit tests passed by the generated code, normalized by the total number of tests. TTFR captures the number of *thought tokens* generated before the first user-visible response (i.e., the initial plan). While token count does not map linearly to wall-clock time, TTFR provides a model-agnostic measure of the delay before any useful surfaced output under a fixed decoding setup. The T/P ratio quantifies overall token efficiency as the average number of tokens generated per problem, including both internal reasoning and user-facing outputs. For interleaved traces, we further assess intermediate response quality using an LLM-as-a-judge (Qwen2.5-7B-Instruct), which labels each segment as *helpful* or *not helpful*.

For inference-time strategies, TTFR is measured at the first generated plan, even if it is later rejected by the autorater, and T/P includes all thought tokens from additional plan generations. Best-of-$N$ plans are generated in parallel, so TTFR is bounded by the slowest plan; R&R evaluates plans sequentially, but each discarded plan is short relative to a full rollout.

*Table 1.* **Interleaved reasoning generalizes beyond coding tasks, lowering TTFR and preserving accuracy.** Plan-first decoding (*Plan-Answer*) substantially lowers time-to-first-response (TTFR) while maintaining or improving pass@1 accuracy compared to the standard *Think-Answer* baseline with a token budget of 4096. Despite being post-trained only on coding data, the learned interleaved behavior transfers effectively to math, text-to-SQL, and reading comprehension tasks.

| Method | MATH500 | | MBPP | | Text-to-SQL | | QuaLITY | | Average | |
|---|---|---|---|---|---|---|---|---|---|---|
| | P@1↑ | TTFR↓ | P@1↑ | TTFR↓ | P@1↑ | TTFR↓ | P@1↑ | TTFR↓ | P@1↑ | TTFR↓ |
| QWEN3-4B | | | | | | | | | | |
| No Thinking | 81.6 | 0 | 48.2 | 0 | 18.6 | 0 | 61.0 | 0 | 52.4 | 0 |
| Think-Answer | 82.8 | 1492 | 50.8 | 1298 | 25.4 | 1542 | 68.4 | 1345 | 56.9 | 1419 |
| **Plan-Answer** | **84.4** | **628** | **52.6** | **523** | **26.0** | **484** | **72.2** | **428** | **58.8** | **516** |
| QWEN3-8B | | | | | | | | | | |
| No Thinking | 82.4 | 0 | 51.4 | 0 | 21.0 | 0 | 64.2 | 0 | 54.8 | 0 |
| Think-Answer | 83.0 | 1587 | 54.5 | 1417 | 28.0 | 1684 | 70.6 | 1463 | 59.0 | 1538 |
| **Plan-Answer** | **85.2** | **625** | **56.8** | **554** | **28.0** | **457** | **76.4** | **443** | **61.6** | **520** |

*Table 2.* **Inference-time strategies:** Plan-level feedback provides earlier and more informative supervision than answer-level feedback, yielding higher accuracy and lower TTFR. Both Best-of-N and R&R (Plan) improve over Plan-Answer without additional training.

| Method | MATH500 | | MBPP | | Text-to-SQL | | QuaLITY | | Average | |
|---|---|---|---|---|---|---|---|---|---|---|
| | P@1↑ | TTFR↓ | P@1↑ | TTFR↓ | P@1↑ | TTFR↓ | P@1↑ | TTFR↓ | P@1↑ | TTFR↓ |
| QWEN3-4B | | | | | | | | | | |
| Plan-Answer | 84.4 | 628 | 52.6 | 523 | 26.0 | 484 | 72.2 | 428 | 58.8 | 516 |
| R&R (Answer) | 84.4 | 4079 | 52.4 | 3321 | 31.2 | 2822 | 74.0 | 2453 | 60.5 | 3169 |
| Best-of-N (5) | 86.2 | **523** | **54.6** | 584 | 28.4 | 568 | **75.2** | 547 | 61.1 | 556 |
| **R&R (Plan)** | **86.8** | 578 | 53.8 | **519** | **31.6** | 552 | 75.0 | **534** | **61.8** | **546** |
| QWEN3-8B | | | | | | | | | | |
| Plan-Answer | 85.2 | 625 | 56.8 | 554 | 28.0 | 457 | 76.4 | 443 | 61.6 | 520 |
| R&R (Answer) | 86.0 | 4279 | 56.2 | 3541 | 34.0 | 2957 | 78.2 | 2614 | 63.6 | 3348 |
| Best-of-N (5) | 85.8 | 628 | **58.2** | 641 | 31.0 | 602 | 78.2 | 582 | 63.3 | 613 |
| **R&R (Plan)** | **89.4** | **611** | 57.4 | **543** | 34.0 | 568 | 78.2 | **562** | **64.8** | **571** |

## 5. Results

**Interleaved reasoning generalizes beyond coding, reducing TTFR and improving task accuracy across domains.** Table 1 compares Think-Answer (TA) and Plan-Answer decoding for QWEN3-4B and QWEN3-8B across MATH500, MBPP, Text-to-SQL, and QuaLITY. Models trained to interleave planning with reasoning on coding data transfer effectively to unseen domains, substantially lowering time-to-first-response (TTFR) while maintaining or improving final task performance. For example, on MATH500, TTFR decreases from $1492 \rightarrow 628$ tokens for QWEN3-4B ($82.8 \rightarrow 84.4$ P@1) and from $1587 \rightarrow 625$ tokens for QWEN3-8B ($83.0 \rightarrow 85.2$ P@1), with similar trends on MBPP and QuaLITY where TTFR is reduced by over 60%. The "No Thinking" baseline yields TTFR = 0 because the model directly outputs an answer without generating any intermediate reasoning tokens; this serves as a lower bound on pre-response delay but typically produces less reliable outputs. Despite being post-trained only on coding tasks, the model exhibits consistent improvements on math, text-to-SQL, and reading comprehension, indicating that explicit planning structures learned in one domain promote

more efficient and accurate reasoning in others. The average gains are therefore not driven by a single benchmark: Plan-Answer improves average pass@1 by 1.9 points for QWEN3-4B and 2.6 points for QWEN3-8B, while cutting average TTFR from 1419/1538 tokens to roughly 520 tokens. The largest relative TTFR reductions occur on Text-to-SQL and QuaLITY, suggesting that the benefit is strongest when monolithic reasoning would otherwise spend many tokens before committing to any user-visible structure. This demonstrates that interleaved reasoning not only reduces pre-answer token overhead but also enhances generalization and response quality across diverse reasoning tasks. Qualitative generated plan and answer examples are provided in Section A.9.

**Early plan-level feedback guides reasoning toward correct solution paths, yielding up to +2–3% higher task accuracy with $7\times$ lower TTFR.** Table 2 compares inference-time control strategies on the RL-trained interleaved model. Unlike answer-level feedback that arrives only after full reasoning, plan-level evaluation intervenes *before* subsequent reasoning begins, allowing the model to revise faulty plans and steer downstream thoughts toward more accu-

*Table 3.* **Token-budget ablation:** Interleaved inference achieves near-saturated accuracy at 4K tokens, highlighting efficient use of available context. While Think-Answer continues to improve with longer reasoning—reflecting redundant token usage—Plan-Answer benefits modestly from larger budgets but remains below Rewind-and-Repeat, indicating stable yet non-adaptive scaling. Rewind-and-Repeat achieves the strongest token efficiency and overall accuracy.

| Method (4096 → 8192) | MATH500 | MBPP | Text-to-SQL | QuaLITY | Average |
|---|---|---|---|---|---|
| QWEN3-4B | | | | | |
| Think-Answer (TA) | 82.8 → 84.2 | 50.8 → 52.0 | 25.4 → 28.2 | 68.4 → 74.6 | 56.9 → 59.8 |
| Plan-Answer | 84.4 → 86.0 | 52.6 → 53.5 | 26.0 → 29.5 | 72.2 → 74.8 | 58.8 → 60.9 |
| R&R (Plan) | 86.8 → 87.2 | 53.8 → 54.3 | 31.6 → 31.7 | 75.0 → 75.4 | **61.8 → 62.2** |
| QWEN3-8B | | | | | |
| Think-Answer (TA) | 83.0 → 88.2 | 54.5 → 56.4 | 28.0 → 31.0 | 70.6 → 78.4 | 59.0 → 63.5 |
| Plan-Answer | 85.2 → 88.8 | 56.8 → 58.5 | 28.0 → 32.5 | 76.4 → 78.2 | 61.6 → 64.5 |
| R&R (Plan) | 89.4 → 89.6 | 57.4 → 59.1 | 34.0 → 34.3 | 78.2 → 78.4 | **64.8 → 65.4** |

*Table 4.* **Rewind statistics.** Fraction of prompts requiring plan rewinds and corresponding approval rates. On average, only 22% of plans require a rewind, and over 80% of corrections succeed within two attempts, demonstrating the efficiency of plan-level R&R.

| Metric | MATH500 | MBPP | Text-to-SQL | QuaLITY | Average |
|---|---|---|---|---|---|
| Plans needing rewind | 120/500 | 55/257 | 60/250 | 40/250 | 22.4% |
| First rewind approved | 60.8% | 54.5% | 63.3% | 67.5% | 61.5% |
| Second rewind approved | 25.8% | 18.2% | 21.7% | 17.5% | 20.8% |
| Rewound P@1 | 86.6% | 72.7% | 85.0% | 82.0% | 81.6% |

rate outcomes. Both Best-of-$N$ and Rewind-and-Repeat (R&R) at the plan stage outperform the base Plan-Answer decoding, showing that lightweight, inference-time guidance alone improves reasoning quality without retraining. For QWEN3-4B, R&R (Plan) boosts MATH500 accuracy from $84.4 \rightarrow 86.8$ while maintaining $\sim 578$ TTFR tokens, over $7\times$ faster than R&R (Answer). Similarly, QWEN3-8B achieves $+2.2$ P@1 on MATH500 and $+0.6$ on MBPP with TTFR $\approx 600$. These results demonstrate that plan interleaving provides earlier, denser supervision that improves both efficiency and final task accuracy.

The advantage of plan-level feedback is that it moves the intervention boundary from the end of a full completion to the beginning of the reasoning process. When answer-level R&R rejects a response, all reasoning tokens used to produce that response are discarded. By contrast, plan-level R&R only discards a short plan prefix, and an accepted plan becomes a compact constraint that conditions the rest of the rollout. This explains why plan-level feedback improves accuracy while preserving low TTFR and total token cost: most of the search happens over interpretable plan proposals rather than complete reasoning traces.

**Think-Answer models rely on longer reasoning chains to improve, indicating inefficient token usage, whereas interleaved models attain comparable accuracy with fewer tokens.** Table 3 varies the available context window (4096 → 8192 tokens). While Think-Answer (TA) models show modest gains with larger budgets (e.g., MATH500 $82.8 \rightarrow 84.2$), interleaved models nearly saturate at 4096 tokens ($86.8 \rightarrow 87.2$ for QWEN3-4B; $89.4 \rightarrow 89.6$

for QWEN3-8B). This demonstrates that 4K tokens are sufficient for plan-conditioned inference, even without extended "thinking" budgets.

In contrast, the TA baseline exhibits poor token efficiency. Its improvements come primarily from longer reasoning sequences, much of which is spent on redundant or self-corrective thought. The tokens-per-problem (T/P) statistics show the same pattern: for QWEN3-8B, TA with an 8192-token budget uses 3025.6 T/P on QuaLITY and 3214.6 T/P on Text-to-SQL, whereas the best plan-based settings use 1908.2 and 1806.2 T/P, respectively, while maintaining or improving accuracy. Empirically, many TA generations fail to terminate naturally and in these cases, the model continues reasoning until truncated, after which a final answer must be explicitly prompted. Interleaved models, by comparison, allocate tokens adaptively and terminate after a few reasoning–answer alternations, effectively regularizing total reasoning length.

In the *Rewind-and-Repeat* setting, this difference becomes particularly salient. Because interleaved models surface an explicit plan before full reasoning, each rewind operates over a concise, interpretable intermediate representation rather than an entire free-form completion. This allows the LLM-as-a-judge to prune low-quality trajectories early — avoiding wasted computation and ensuring that subsequent reasoning unfolds around a verified plan, whereas TA models can only rewind after full generations, offering no opportunity for early correction. On MATH500, for example, the TA model frequently degenerates into repetitive or self-contradictory reasoning and fails to recover once diverged,

whereas PLANTAIN avoids this failure mode by grounding the process through explicit subgoals that constrain and stabilize subsequent thoughts. Overall, these results show that interleaved inference regularizes reasoning depth, avoids unnecessary computation, and achieves higher token efficiency under fixed or limited context budgets. See Section A.10 for qualitative traces illustrating these dynamics.

This qualitative behavior clarifies why the aggregate token-budget results favor plan-conditioned inference. Extra thinking tokens help TA only when the model can use them to repair a trajectory, but in failure cases the additional budget can instead amplify repeated checks or inconsistent partial derivations. PLANTAIN exposes the trajectory choice earlier: once the plan is accepted, subsequent reasoning is anchored to a smaller set of explicit subgoals, and when the plan is rejected, the correction happens before the model has committed to a long chain of downstream reasoning.

**A small number of plan-level interventions recover most failure cases, highlighting the efficiency of structured feedback.** Table 4 analyzes how often plan-level intervention is needed and how quickly the model converges under the Rewind & Repeat (R&R) strategy. Across all benchmarks, only 20–25% of plans require a rewind, and the majority of those are corrected on the first retry (60–65%). A small fraction (15–25%) benefit from a second rewind, after which success rates exceed 80%. These results indicate that most initial plans are already well-formed and R&R primarily serves to re-ground reasoning early, preventing the model from pursuing unproductive solution paths. This mechanism contributes directly to the token efficiency observed in Table 2, where plan-level feedback reduces reasoning depth without sacrificing accuracy. Qualitatively, we find that rewinds promote more stable and targeted reasoning trajectories, allowing the model to converge faster on valid solutions with minimal additional tokens.

## 6. Related Work

**LLM Overthinking and Adaptive Thinking.** Reasoning models such as OpenAI O1 (Jaech et al., 2024) and DeepSeekR1 (Guo et al., 2025) are trained using RL-based methods to improve task accuracy on complex logic and reasoning tasks. Such LLMs notoriously suffer from an overthinking phenomenon where they produce verbose and redundant reasoning steps (Chen et al., 2024; Sui et al., 2025), which wastes time, risks exceeding the token budget, and in the extreme, can introduce logical fallacies that lead to incorrect final answers (Sui et al., 2025).

One proposed solution is to incorporate a thought length penalty into the RL objective, penalizing verbose and incorrect answers while encouraging the model to produce more concise reasoning steps (Arora & Zanette, 2025; Yeo

et al., 2025; Luo et al., 2025). Another strategy focuses on data curation, where variable length Chain-of-Thought (CoT) reasoning datasets are constructed for Supervised Fine-tuning (SFT) (Kang et al., 2025). This is achieved by using heuristics or another LLM to summarize and compress longer reasoning chains without losing key information. Finally, Munkhbat et al. (2025) use best-of-N sampling to generate more concise reasoning paths for data curation. A key difference from all of these approaches is that we do not wish to penalize long reasoning chains *per se*, but rather to improve interactivity and ensure that reasoning effort is well spent. We show that training for interleaved reasoning naturally encourages concise reasoning without explicit length penalties, while allowing users to ensure that reasoning proceeds from the correct premises.

**Post-hoc Reranking and Inference-Time Steering.** We consider inference-time techniques that exploit plan proposals to improve generation quality. This is related to the broader literature on inference-time scaling and aggregation, which is too large to properly survey here. We draw particular inspiration from early approaches, such as contrastive decoding (Li et al., 2022) and energy-based reranking models (Bakhtin et al., 2021), which generate multiple candidate continuations and select those that best satisfy predefined constraints. RANKGEN (Krishna et al., 2022) introduces a large encoder trained with contrastive objectives to assess prefix–continuation compatibility, improving coherence and topical relevance when applied to candidate selection. RankRAG (Yu et al., 2024) unifies reranking and answer generation in retrieval-augmented settings. More recently, AGGLM (Zhao et al., 2025) merges reasoning outputs from several trajectories into a unified answer through inference-time aggregation.

While these methods can improve performance, they require producing and filtering many full reasoning traces. In contrast, our inference-time steering techniques are more efficient because they intervene directly within the decoding process itself, by producing and evaluating *partial* generations before continuing with the remainder of the response. Conceptually, this means the unit of generation for evaluation is smaller: instead of scoring entire completions post-hoc, as in RANKGEN or AGGLM, our approach incorporates plan assessment mid-generation. Unlike speculative decoding (Leviathan et al., 2023), this does not require interrupting the decoding process or maintaining multiple concurrent hypotheses. The interleaved model naturally alternates between planning, reasoning, and answering within a single forward pass. This shifts the intervention from post-hoc reranking to real-time guidance, enabling efficiency gains without auxiliary reranking stages.

**Plan Guided Reasoning.** The ReAct framework interleaves chain-of-thought reasoning with actions allowing the model

to incorporate external information sources through tool calls, thereby grounding the model and reducing hallucinations in the final answer (Yao et al., 2023). A particularly relevant follow-up is Pre-Act, which creates a multi-step execution plan and reasoning for each action (Rawat et al., 2025). In contrast to ReAct-style frameworks, PLANTAIN produces intermediate answers that are surfaced to the user allowing them to intervene and correct the model's reasoning paths. Similar to ReAct, we observe that grounding the reasoning in an explicit plan leads to improved downstream accuracy. Other prior works have shown that plan-based models, sometimes referred to as *blueprints*, help with model robustness and attribution (Huot et al., 2023; Gurung & Lapata, 2025; Fierro et al., 2024).

**Interactive LLM Interfaces and Decoding Control.** User-facing LLM interfaces, such as OpenAI's Deep Research, Gemini's multi-step reasoning mode, and most standard chat UIs allow users to stop generation mid-thought, truncate rambling explanations, or redirect the model before a full chain of thought is completed (OpenAI, 2025). While effective for improving user-facing responsiveness and control, UI-level interruption does not influence the model's internal reasoning policy. As a result, the model continues to plan to overthink, even if the user frequently stops it. From a learning perspective, the system never receives a training signal that shorter or more structured reasoning is preferred. In contrast, interleaved reasoning modifies the generation process itself, prompting the model to surface compact, high-value intermediate outputs by default while continuing internal thought as needed.

## 7. Conclusion

We introduced PLANTAIN, a post-training framework that elicits *interleaved reasoning* behavior in large reasoning models. By encouraging models to first produce an explicit plan and then alternate between internal reasoning and user-facing responses, PLANTAIN reduces time-to-first-response (TTFR) and improves controllability. Empirically, interleaved reasoning achieves up to $7\times$ lower TTFR and $+2$–$3\%$ higher task accuracy compared to standard *Think-Answer* decoding, while improving token efficiency on several tasks. Our interleaved model learns to structure its reasoning process naturally, without explicit length penalties or handcrafted constraints. At inference time, we introduced two plan-level control strategies, Best-of-$N$ selection and Rewind & Repeat, that further improve response quality and efficiency without additional training. Although trained solely on code generation tasks, the resulting model generalizes effectively to broader reasoning domains such as math and long-context question answering.

The central claim is not that surfaced plans are always sufficient for solving hard problems, but that post-training can create a useful point of control before the model commits to long-form reasoning. This distinction is important for interpreting TTFR: we measure the number of hidden tokens before the first useful visible response, rather than directly measuring human-perceived latency. Within this controlled setting, the results show that making the plan visible enables cheaper feedback and more targeted reasoning, while preserving the ability to continue with internal computation when needed.

PLANTAIN focuses on structured, verifiable tasks where plan correctness can be automatically assessed, and our experiments use 4B and 8B open models because RL post-training substantially larger reasoning models is expensive. An important next step is to conduct human studies evaluating the perceived usefulness, interpretability, and responsiveness of interleaved reasoning in interactive settings. We also plan to extend evaluations to agentic tool-use benchmarks and harder reasoning benchmarks such as AIME or GPQA, where early plan inspection may be especially valuable because later actions or long reasoning paths are costly to undo. Additional ablations should isolate the contributions of SFT and RL post-training, since we expect SFT to teach the plan-first format while PPO improves task performance under verifiable rewards. Finally, our plan feedback experiments use a fixed LLM-as-a-judge user simulator; broader robustness studies with alternative judges, human feedback, or lightweight task-specific verifiers would better characterize how sensitive PLANTAIN is to the source of plan feedback.

## Acknowledgments

We are deeply grateful to Ting-yun Chang, Frederick Zhang, and Ming Zhong, Alexandra Chronopoulou, Xiang Zhou, and Shyam Upadhyay for the discussions and thoughtful feedback that helped refine the methodology, experimental design, and paper.

## Impact Statement

This paper describes fundamental research on techniques for inference-time guidance in reasoning models, specifically through interleaved reasoning where models alternate between internal thought and visible intermediate plans. The primary intended use case that we analyze in this work is interactive problem solving, where reducing the delay before a useful surfaced response and enabling user intervention are core challenges. Surfacing intermediate plans can help make AI systems easier to use, correct, and monitor while solving complex tasks in areas such as coding and math. We do not speculate about broader impacts that may follow from this technical contribution.

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

# A. Appendix

## A.1. PLANTAIN Algorithm

---

**Algorithm 4** PLANTAIN: Post-training to Elicit Interleaved Reasoning

---

**Input:** Base LM $\pi_{\theta_0}$; reference policy $\pi_{\text{ref}}$; value net $V_\phi$; prompts $\mathcal{X}$; SI template $\texttt{SI}_{\text{plan-first}}$; synthetic generator $\Pi^\star$ (larger LM); weights $\alpha_{\text{fmt}}, \alpha_{\text{corr}}, \alpha_{\text{help}}, \alpha_{\text{unit}}$; PPO coeff $\beta$

**Output:** Interleaved model $\pi_\theta$ (plan-first, then alternating thoughts/answers)

    **Synthetic Interleaved Trace Dataset**

1: $\mathcal{D}_{\text{interleave}} \leftarrow \varnothing$
2: **for all** $x \in \mathcal{X}$ **do**
3:     $x' \leftarrow \texttt{SI}_{\text{plan-first}}(x)$
4:     $\tau = (t_1, a_1, t_2, a_2, \ldots, t_n, a_n) \sim \Pi^\star(\cdot \mid x')$ {$a_1$ is the explicit *plan*}
5:     $\mathcal{D}_{\text{interleave}} \leftarrow \mathcal{D}_{\text{interleave}} \cup \{(x', \tau)\}$
6: **end for**
    **Supervised Fine-tuning**
7: $\theta \leftarrow \arg\min_\theta \sum_{(x', \tau) \in \mathcal{D}_{\text{interleave}}} \big( - \log \pi_\theta(\tau \mid x')\big)$ {Shift style from monolithic to interleaved}
    **RL Post-Training (PPO)**
8: **repeat**
9:     Sample $(x', \cdot) \sim \mathcal{D}_{\text{interleave}}$; rollout $y \sim \pi_\theta(\cdot \mid x')$
10:     $r_{\text{fmt}} \leftarrow \text{FORMATOK}(y)$ {plan-first; valid interleaving; required sections}
11:     $r_{\text{corr}} \leftarrow \text{UNITTESTPASSRATE}(x', y)$
12:     $r_{\text{help}} \leftarrow \text{LLMJUDGEHELPFULNESS}(x', y)$
13:     $r_{\text{unit}} \leftarrow \mathbf{1}\{\text{unit\_tests present in } y\}$
14:     $g \leftarrow \mathbf{1}\{r_{\text{fmt}} = 1\}$ {gate downstream rewards on correct interleaved format}
15:     $r(x', y) \leftarrow \alpha_{\text{fmt}} r_{\text{fmt}} + g \cdot \big(\alpha_{\text{corr}} r_{\text{corr}} + \alpha_{\text{help}} r_{\text{help}} + \alpha_{\text{unit}} r_{\text{unit}}\big)$
16:     Compute advantages $\hat{A}$ with GAE using $V_\phi$; update $\theta, \phi$ via PPO:
17:     $\max_\theta \mathbb{E}\left[\text{clip}\left(\frac{\pi_\theta}{\pi_{\theta_{\text{old}}}}, 1 \pm \epsilon\right) \hat{A}\right] - \beta \, \text{D}_{\text{KL}}\big(\pi_\theta(\cdot \mid x') \,\|\, \pi_{\text{ref}}(\cdot \mid x')\big)$
18: **until** convergence
19: **return** $\pi_\theta$

---

**Algorithm 5** Inference Strategy 1: Best-of-$N$ Plan

---

**Input:** Trained interleaved model $\pi_\theta$; prompt $p$; temperature $\tau > 1$; number of plans $N$; LLM Autorater $R_{\text{select}}$

1: Form plan-first instruction $p' \leftarrow \texttt{SI}_{\text{plan-first}}(p)$
2: Generate candidate plans $\mathcal{A} \leftarrow \{a_i \sim \pi_\theta(a \mid p'; \tau)\}_{i=1}^N$
3: Select $i^\star \leftarrow R_{\text{select}}(p, \mathcal{A})$;    $a^\star \leftarrow a_{i^\star}$
4: Roll out remainder conditioned on $a^\star$: $y \sim \pi_\theta(\cdot \mid p', a^\star)$
5: **return** $y$

---

**Algorithm 6** Inference Strategy 2: Rewind & Repeat - Iterative Plan Rejection Sampling

---

**Input:** Trained interleaved model $\pi_\theta$; prompt $p$; maximum retries $T$; LLM Autorater $R_{\text{bin}}$

1: $p' \leftarrow \texttt{SI}_{\text{plan-first}}(p)$;    $\mathcal{H} \leftarrow \varnothing$
2: **for** $t = 1$ to $T$ **do**
3:     $c_t \leftarrow \text{AUGMENT}(p', \mathcal{H})$
4:     Propose plan $a_t \sim \pi_\theta(a \mid c_t)$
5:     $d_t \leftarrow R_{\text{bin}}(p, a_t)$
6:     **if** $d_t = \texttt{accept}$ **then**
7:         **return** $y \sim \pi_\theta(\cdot \mid c_t, a_t)$
8:     **else**
9:         $\mathcal{H} \leftarrow \mathcal{H} \cup \{a_t\}$
10:     **end if**
11: **end for**
12: Sample a random plan $a_r \sim \text{Uniform}(\mathcal{H})$
13: **return** $y \sim \pi_\theta(\cdot \mid p', a_r)$

## A.2. Interleaved Reasoning System Instruction

---

**Interleaved Reasoning System Prompt**

**System.** You are a helpful assistant. You reason through problems step-by-step before providing an answer. You conduct your reasoning within `<think></think>` and share partial answers that are useful for the user within `<answer></answer>`. You continue this pattern of `<think>...</think>` `<answer>...</answer>` `<think>...</think>` `<answer>...</answer>` until you reach the final answer.

**User.** {`insert prompt here`}
**Assistant.**

---

## A.3. LLM Plan Autorater

---

**Plan Autorater Prompt**

**Role.** You are an expert plan evaluator. Given a prompt and a single plan, your task is to determine if the plan is good enough to proceed with.

**Evaluation criteria.**

1. Completeness: Does the plan address all aspects of the prompt?
2. Feasibility: Is the plan realistic and implementable?
3. Clarity: Is the plan clear and well-structured?
4. Alignment: Does the plan align with the user's intent?

Carefully evaluate the plan based on the criteria above. Consider whether this plan provides a solid foundation for addressing the user's request.

**Approval rubric.**
A plan is APPROVED (`Decision: TRUE`) if it:

1. Directly addresses the core question or request
2. Provides clear, actionable steps
3. Covers the essential aspects without being overly complex
4. Aligns with the user's apparent intent

**Rejection rubric.**
A plan is REJECTED (`Decision: FALSE`) if it:

1. Misses key aspects of the request
2. Is too vague or abstract to act on
3. Contains unrealistic or impractical elements
4. Is incomplete or poorly structured

**Decision format.**
Respond with exactly one line in this format:

    `Decision: TRUE` or `Decision: FALSE`

Please proceed with the evaluation. `Decision:`

---

**Best-of-$N$ Plan Selection Prompt**

**Role.** You are an expert plan evaluator. Given a prompt and several candidate plans, select the single plan that provides the best foundation for answering the prompt correctly.

**Evaluation criteria.** Choose the plan that is most complete, feasible, clear, and aligned with the user's intent. Prefer plans that address all important constraints in the prompt and are likely to lead to a correct final response.

**Decision format.** Respond with exactly one line containing the selected plan index:

         `Selected: 1`

**Prompt:** {insert prompt here}
**Candidate plans:** {insert indexed candidate plans here}
`Selected:`

---

## A.4. Dataset Details

| Dataset | Size |
|---|---|
| BigCodeBench (Zhuo et al., 2024) | 50 |
| MBPP (Austin et al., 2021) (Concat-2) | 50 |
| MATH500 (Lightman et al., 2023) (Concat-2) | 50 |
| Multiple Solutions | 26 |

*Table 5.* SFT dataset details.

| Dataset | Size |
|---|---|
| BigCodeBench (Zhuo et al., 2024) | 500 |
| MBPP (Austin et al., 2021) (Concat-2) | 50 |
| MATH500 (Lightman et al., 2023) (Concat-2) | 50 |
| Multiple Solutions | 26 |

*Table 6.* RL dataset details.

## A.5. Training and Evaluation Datasets

**SFT and RL Training Datasets.** We use a combination of coding and mathematical reasoning datasets to train our interleaved reasoning models.

1. **BigCodeBench (BCB)** (Zhuo et al., 2024) is a benchmark of 1,140 Python programming tasks that require diverse function calls from common libraries such as `numpy` and `matplotlib`. Each task includes an average of 5.6 unit tests with 99% branch coverage. For training, we split the dataset into train and test sets, and randomly sample 50 prompts from the training split to generate synthetic interleaved responses. To better suit interleaved reasoning, we modify the original prompts to include not only code generation but also a brief solution outline and associated unit tests.
2. **Mostly Basic Python Programs (MBPP)** (Austin et al., 2021) consists of 974 crowd-sourced Python problems designed to be solvable by entry-level programmers. Each problem includes a short text description and three test cases. To promote multi-step reasoning, we sample multiple problems and concatenate them into a single composite prompt that requires the model to solve each sequentially.
3. **MATH500** (Lightman et al., 2023) contains 500 diverse math problems spanning topics such as probability, algebra, trigonometry, and geometry. Similar to MBPP, we randomly combine multiple problems into a single prompt to encourage multi-stage reasoning and plan refinement.

**Evaluation Datasets.** We evaluate the trained models across domains that require coding, mathematical reasoning, symbolic translation, and long-context comprehension.

- **BigCodeBench (BCB)** (Zhuo et al., 2024): challenging Python coding prompts requiring composition across multiple libraries.
- **Mostly Basic Python Programs (MBPP)** (Austin et al., 2021): simple Python programming tasks solvable by entry-level programmers.
- **MATH500** (Lightman et al., 2023): diverse mathematical reasoning problems across algebra, probability, and geometry.
- **BirdSQL** (Li et al., 2023): cross-domain text-to-SQL benchmark with over 12k question–SQL pairs across 95 databases.
- **QuaLITY** (Pang et al., 2021): long-document multiple-choice QA benchmark designed to test reasoning over extended contexts.

### A.6. Training Details

| Parameter | Value |
|---|---|
| Actor learning rate | $1 \times 10^{-6}$ |
| Critic learning rate | $1 \times 10^{-6}$ |
| Train batch size per gpu | 32 |
| Validation batch size | 256 |
| PPO mini batch size | 32 |
| PPO micro batch size | 16 |
| Critic micro batch size | 8 |
| KL coefficient | 0.001 |
| KL loss type | low variance KL |
| Max prompt length | 3096 tokens |
| Max response length | 2500 tokens |
| Sampling temperature | 0.7 |
| Number of samples per prompt | 8 |
| Stable training threshold ($\epsilon$) | 0.05 |
| Critic warmup steps | 0 |
| Evaluation frequency | 50 steps |
| Tensor model parallel size | 2 |

*Table 7.* Training hyperparameters.

Experiments were conducted building on VERL (Sheng et al., 2024), an efficient reinforcement learning framework for post-training language models. We performed all experiments on 8 NVIDIA H100 GPUs with 80GB memory in a Google Cloud VM. We also used a consistent set of hyperparameters to ensure fair comparison between methods. We evaluate and save every 50 steps during training, and continue training from the last saved checkpoint if the training is interrupted (e.g., OOM).

### A.7. Synthetic Interleaved Response Generation

We generate synthetic interleaved reasoning traces by iteratively prompting a larger model (Qwen3-32B). For Big-

CodeBench, we modify the vanilla coding prompts, to also ask for a solution outline and unit tests in addition to the code solution. We prompt the model to generate thought and intermediate response traces in this order: `thought → code outline → thought → code solution → thought → unit tests`. To prevent the base model from spending all its output budget on thinking, we terminate the thought generate after $N = 256$ tokens, clean up the final sentence, and append a `</think>` token to mark the end of a subthought. Then we reprompt the model provided the previously generated thoughts as context to generate the next intermediate response. We provide an example of this interleaved trace at Example A.8 below.

### A.8. Training Dataset Prompt Examples

---
**BCB-Outline-Code-Unit**

**Question:**
Save the provided Pandas DataFrame "df" in a pickle file with the given name, read it back for validation, and delete the intermediate file. The function should output with: `loaded_df (pd.DataFrame)`: The loaded DataFrame from the specified file. You should write self-contained code starting with:

```python
import pickle
import os

def task_func(df,
    file_name="save.pkl"):
    # function implementation
    pass
```

First, outline the solution in a markdown format. Then, write the code to implement the solution. Finally, generate unit tests to test the code. Format the unit tests as a python function with a docstring. Use this exact format:

```python
import unittest
from task_func import task_func
class Test(unittest.TestCase):
    def test_case_1(self):
        # Test case 1 description
        result = task_func(...)
        self.assertEqual(result,
                expected_value)
```

---
**MATH500 Concatenate 2**

**Question:**
Solve the following problems:
1) Find all $p$ which satisfy both the inequalities $0 \geq 54p - 144$ and $0 > 12 - 20p$. Express your answer in interval notation, reducing any fractions in your answer.

---

2) A strictly increasing sequence of positive integers $a_1$, $a_2$, $a_3$, ... has the property that for every positive integer $k$, the subsequence $a_{2k-1}$, $a_{2k}$, $a_{2k+1}$ is geometric and the subsequence $a_{2k}$, $a_{2k+1}$, $a_{2k+2}$ is arithmetic. Suppose that $a_{13} = 2016$. Find $a_1$.

**Answer:**

<think> Okay, so I need to find all values of p that satisfy both inequalities: $0 \geq 54p - 144$ and $0 > 12 - 20p$. Hmm, let me start by solving each inequality separately and then find their intersection ... </think>

<answer> $\left(\frac{3}{5}, \frac{8}{3}\right]$ </answer>

<think> Okay, so I need to solve this problem about a strictly increasing sequence of positive integers. The sequence has some properties related to geometric and arithmetic subsequences. ... </think>

<answer> 504 </answer>

---

### MBPP Concatenate 2

**Question:**

Solve the following coding problems:
1) Write a function to reverse words in a given string.
2) Write a function that matches a string that has an a followed by zero or more b's by using regex.
You should write self-contained code for each problem starting with: `def task_func(args):`

**Answer:**

<think> Okay, I need to write a function that reverses the words in a given string. Let me think about how to approach this. So, the input is a string like "Hello world", and the output should be "world Hello"... </think>

<answer>

```python
def task_func(s):
    return '
    ↪    '.join(reversed(s.split()))
```

</answer>

<think> Okay, I need to write a function that checks if a string has an 'a' followed by zero or more 'b's using regex. Let me think about how to approach this. First, I remember that regular expressions use certain patterns. The problem says "a followed by zero or more b's". So the 'a' must be there, and then after it, there can be any number of 'b's, including none... </think>

<answer>

```python
import re
def task_func(text):
    patterns = 'ab*?'
    if re.search(patterns, text):
            return ('Found a match!')
    else:
            return ('Not matched!')
```

</answer>

---

### Multiple Code Solutions

**Question:**

---

Implement a function that performs basic string compression (e.g., 'aabcccccaaa' becomes 'a2b1c5a3'). Provide an iterative solution and one using itertools.
You should write self-contained code for each problem starting with: `def task_func(args):`

**Answer:**

<think> Okay, I need to write an iterative Python function for string compression. Let's see. The problem says that consecutive repeated characters should be replaced by the character followed by the count. And if the compressed string isn't shorter, return the original. First, I should think about how to process the string. Since it's iterative, I can't use recursion... </think>

<answer>

```python
def task_func(s: str) -> str:
    if not s:
        return s

    compressed = []
    current_char = s[0]
    count = 1

    for char in s[1:]:
        if char == current_char:
            count += 1
        else:
            compressed.append(
                current_char + str(count)
            )
            current_char = char
            count = 1

    # Append the last character
    # and its count
    compressed.append(current_char
            + str(count))

    compressed_str =
    ↪    ''.join(compressed)
    return compressed_str
        if len(compressed_str) <
        ↪    len(s)
        else s
```

</answer>

<think> Okay, I need to write a Python function called task_func that compresses a string using itertools. The goal is to replace consecutive repeated characters with the character followed by the count. And if the compressed version isn't shorter, return the original. First, I remember that itertools has groupby, which groups consecutive elements ... </think>

<answer>

```python
from itertools import groupby

def task_func(s: str) -> str:
    compressed = ''.join(char
        + str(len(list(group)))
    for char, group in groupby(s))
    return compressed
        if len(compressed) < len(s)
        else s
```

</answer>

## A.9. Generated Plan and Answers

### MATH500

**Question:**
Convert the point $(0, 3)$ in rectangular coordinates to polar coordinates. Enter your answer in the form $(r, \theta)$, where $r > 0$ and $0 \leq \theta < 2\pi$.

**Plan:**

1. Identify the rectangular coordinates $(x, y) = (0, 3)$.
2. Use the formula $r = \sqrt{x^2 + y^2}$ to calculate the radial distance $r$.
3. Determine the angle $\theta$ using the relationship $\theta = \arctan\left(\frac{y}{x}\right)$, adjusting for the correct quadrant based on the signs of $x$ and $y$.
4. For the point $(0, 3)$, recognize that it lies on the positive $y$-axis, so $\theta = \frac{\pi}{2}$.
5. Verify that $\theta$ is within the required range $0 \leq \theta < 2\pi$.
6. Write the final polar coordinates in the form $(r, \theta)$.

**Answer:** $(3, \frac{\pi}{2})$

### BirdSQL

**Database: debit_card_specializing**

**Table: customers**

| | |
|---|---|
| CustomerID | identification of the customer (integer) |
| Segment | client segment (text) |
| Currency | Currency (text) |

**Table: gasstations**

| | |
|---|---|
| GasStationID | Gas Station ID (integer) |
| ChainID | Chain ID (integer) |
| Country | No description (text) |
| Segment | chain segment (text) |

**Table: products**

| | |
|---|---|
| ProductID | Product ID (integer) |
| Description | Description (text) |

**Table: transactions_1k**

| | |
|---|---|
| TransactionID | Transaction ID (integer) |
| Date | Date (date) |
| Time | Time (text) |
| CustomerID | Customer ID (integer) |
| CardID | Card ID (integer) |
| GasStationID | Gas Station ID (integer) |
| ProductID | Product ID (integer) |
| Amount | Amount (integer) |
| Price | Price (real) |

*Note: total price = Amount x Price*

**Table: yearmonth**

| | |
|---|---|
| CustomerID | Customer ID (integer) |
| Date | Date (text) |
| Consumption | consumption (real) |

**Question:** In 2012, who had the least consumption in LAM?

**Plan:**

1. Filter the `yearmonth` table for the year 2012 by checking if the 'Date' starts with '2012'.
2. Join the `yearmonth` table with the `gasstations` table using the `GasStationID` to retrieve country information.
3. Filter the joined dataset to include only rows where `Country` is `LAM`.
4. Aggregate the `Consumption` values per customer using `GROUP BY CustomerID`.
5. Sort the results in ascending order of the total consumption.
6. Use `LIMIT 1` to retrieve the customer with the least consumption.

**Answer:**

```
SELECT y.CustomerID,
↪   SUM(y.Consumption)
    AS total_consumption
FROM yearmonth y
JOIN gasstations g
    ON y.GasStationID =
    ↪   g.GasStationID
WHERE LEFT(y.Date, 4) = '2012'
    AND g.Country = 'LAM'
GROUP BY y.CustomerID
ORDER BY total_consumption ASC
LIMIT 1;
```

## A.10. MATH500 Think-Answer vs. Rewind-and-Repeat

### MATH500 Problem 153 Rewind-and-Repeat

**Question:** The medians $AD$, $BE$, and $CF$ of triangle $ABC$ intersect at the centroid $G$. The line through $G$ that is parallel to $BC$ intersects $AB$ and $AC$ at $M$ and $N$, respectively. If the area of triangle $ABC$ is 144, then find the area of triangle $ENG$.

<think> ... </think>
<answer>

1. Use coordinate geometry to assign coordinates to triangle $ABC$, simplifying calculations.
2. Determine the coordinates of the centroid $G$ using the midpoint formula.
3. Find the equation of the line through $G$ that is parallel to $BC$.
4. Calculate the coordinates of points $M$ and $N$, where this line intersects $AB$ and $AC$, respectively.
5. Use the coordinates of points $E$, $N$, and $G$ to compute the area of triangle $ENG$ using the shoelace formula or determinant method.
6. Relate the area of triangle $ENG$ to the area of triangle $ABC$, which is given as 144, to find the numerical value.

</answer>
<think> Okay, let's start by assigning coordinates to triangle ABC. I'll place point A at $(0, 0)$, B at $(2b, 0)$, and C at $(0, 2c)$. This makes AB and AC the axes, simplifying calculations. The centroid G is the average of the coordinates, so G should be at $(\frac{2b}{3}, \frac{2c}{3})$.

Next, I need the equation of the line through G parallel to BC. The slope of BC is $\frac{2c-0}{0-2b} = -\frac{c}{b}$. So the line through

G with this slope is $y - \frac{2c}{3} = (-\frac{c}{b})(x - \frac{2b}{3})$. Let me rearrange this equation.

*(Intermediate reasoning steps omitted for brevity.)*

So triangle ENG has points $E(0, c)$, $N(0, \frac{4c}{3})$, and $G(\frac{2b}{3}, \frac{2c}{3})$. Let's compute its area. Using the shoelace formula:

Area $= \frac{1}{2}|(x_E(y_N - y_G) + x_N(y_G - y_E) + x_G(y_E - y_N))|$.

Plugging in:

$x_E = 0, y_E = c; x_N = 0, y_N = \frac{4c}{3}; x_G = \frac{2b}{3}, y_G = \frac{2c}{3}$.

Calculating each term:

First term: $0 \cdot (\frac{4c}{3} - \frac{2c}{3}) = 0 \cdot (\frac{2c}{3}) = 0$.

Second term: $0 \cdot (\frac{2c}{3} - c) = 0 \cdot (-\frac{c}{3}) = 0$.

Third term: $(\frac{2b}{3}) \cdot (c - \frac{4c}{3}) = (\frac{2b}{3}) \cdot (-\frac{c}{3}) = -\frac{2bc}{9}$.

Taking absolute value and multiplying by $\frac{1}{2}$: $\frac{1}{2} \cdot |-\frac{2bc}{9}| = \frac{1}{2} \cdot \frac{2bc}{9} = \frac{bc}{9}$.

So the area of ENG is $\frac{bc}{9}$.

Now, relate this to the area of ABC. The area of ABC is 144. Since ABC is a right triangle with legs $2b$ and $2c$, its area is $(\frac{1}{2}) \cdot 2b \cdot 2c = 2bc$. So $2bc = 144 \Rightarrow bc = 72$.

Therefore, the area of ENG is $\frac{72}{9} = 8$.

Wait, that seems straightforward. Let me check each step again. The coordinates assignment, centroid calculation, line equation, points M and N, and area computation. All steps seem correct. The key was setting up the coordinate system and using the shoelace formula. The answer should be 8. </think>

<answer> 8 </answer>

---

## MATH500 Problem 153 Think-Answer

**Question:** The medians $AD$, $BE$, and $CF$ of triangle $ABC$ intersect at the centroid $G$. The line through $G$ that is parallel to $BC$ intersects $AB$ and $AC$ at $M$ and $N$, respectively. If the area of triangle $ABC$ is 144, then find the area of triangle $ENG$.

<think> Okay, so I need to find the area of triangle $ENG$ in triangle $ABC$, where the medians $AD$, $BE$, and $CF$ intersect at centroid $G$. There's a line through $G$ parallel to $BC$ that intersects $AB$ at $M$ and $AC$ at $N$. The area of $ABC$ is 144. Hmm, let me try to visualize this first.

First, let me recall some properties of centroids. The centroid divides each median into a ratio of 2:1, with the longer part being closer to the vertex. So, for example, $AG : GD = 2 : 1$, $BG : GE = 2 : 1$, and $CG : GF = 2 : 1$.

Now, there's a line through $G$ parallel to $BC$. Since $BC$ is the base of the triangle, this line is going to be somewhere inside the triangle. It intersects $AB$ at $M$ and $AC$ at $N$. So, $MN$ is parallel to $BC$ and passes through $G$. I need to figure out where exactly $M$ and $N$ are located on $AB$ and $AC$.

*(Intermediate reasoning steps omitted for brevity.)*

Since $MN$ is parallel to $BC$, triangle $AMN$ is similar to triangle $ABC$ by the Basic Proportionality Theorem or Thales' theorem. Wait, but $MN$ isn't necessarily a midline unless it's halfway. However, since it passes through the centroid, which divides the medians in a 2:1 ratio, maybe I can find the ratio of similarity.

Let me think. Let me consider coordinates. Maybe coordinate geometry can help here. Let me assign coordinates to triangle $ABC$. Let me place point $A$ at $(0, 0)$, $B$ at $(2b, 0)$, and $C$ at $(2c, 2d)$. Wait, but maybe it's better to use coordinates that make calculations easier. Since the centroid divides the medians in 2:1, maybe choosing coordinates where the centroid is at $(0, 0)$? Wait, but maybe that complicates things. Alternatively, let me use standard coordinates.

Alternatively, let me take coordinates such that $BC$ is horizontal for simplicity. Wait, but since $MN$ is parallel to $BC$, maybe it's better to have $BC$ horizontal. Let me try.

First, equation of $AB$: from $A(a, b)$ to $B(0, 0)$. The parametric equations can be written as $x = a - a \cdot t, y = b - b \cdot t$, where $t$ ranges from 0 to 1. Wait, or maybe better to use a parameter $s$.

Alternatively, let me find the equation of $AB$. The slope of $AB$ is $\frac{0-b}{0-a} = \frac{b}{a}$. So equation is $y = (\frac{b}{a})x$.

Similarly, equation of $AC$: from $A(a, b)$ to $C(c, 0)$. The slope is $\frac{0-b}{c-a} = -\frac{b}{c-a}$. Equation is $y - b = [-\frac{b}{c-a}](x - a)$.

Now, the line $y = b/3$ intersects $AB$ and $AC$. Let me find coordinates of $M$ and $N$.

For point $M$ on $AB$: $y = (\frac{b}{a})x = \frac{b}{3}$. Therefore, $x = (\frac{b}{3}) \cdot (\frac{a}{b}) = \frac{a}{3}$. Therefore, coordinates of $M$ are $(\frac{a}{3}, \frac{b}{3})$.

Similarly, for point $N$ on $AC$: Let me solve $y = \frac{b}{3}$ with equation of $AC$.

Equation of $AC$: $y = [-\frac{b}{c-a}](x - a) + b$.

Set $y = b/3$:

$\frac{b}{3} = [-\frac{b}{c-a}](x - a) + b$

...

Wait no. Wait no. Wait no. Wait no... </think>

