# OpenReview forum: "PLANTAIN: Plan-Answer Interleaved Reasoning"
_ICML.cc/2026/Conference — ICML 2026 spotlight_

### Official Review · Reviewer_K9Wz · 2026-03-09

**Soundness:** 3
**Presentation:** 3
**Significance:** 3
**Originality:** 3
**Overall Recommendation:** 4
**Confidence:** 4

**Summary:**

This paper introduces PLANTAIN, a framework designed to replace the standard think-then-answer paradigm in reasoning models with interleaved reasoning. PLANTAIN trains models to output an explicit, step-by-step plan as their first intermediate response before completing the full reasoning process to address the high latency of the previous paradigm. The authors also propose two inference-time strategies: Best-of-N-Plans and Rewind-and-Repeat. The authors report that their framework reduces time-to-first-response by over 60% and achieves an average 6% improvement in pass@1 accuracy across coding and math benchmarks.

**Compliance With Llm Reviewing Policy:**

Affirmed.

**Final Justification:**

Thank authors for the detailed response, which has addressed some of my concerns. Based on this, I will maintain my current rating (weak accept).

**Key Questions For Authors:**

1. The inference strategies used in this paper both rely on an external LLM autorater to evaluate the initial plans. Calling such an LLM judge apparently will cause potential network latency, especially in a real-time deployment scenario. Therefore, has the reported TTFR time taken this into account? If not, what will the actual (real situation) TTFR time be?

2. As mentioned in weakness (1), is it possible to also report the total time usage?

**Limitations:**

Yes

**Strengths And Weaknesses:**

Strengths:

1. The new paradigm introduced by PLANTAIN, combined with its Best-of-N-Plans and Rewind-and-Repeat strategy, shows novelty.

2. The PLANTAIN framework presents a significant improvement compared to traditional think-answer methods, largely reducing the TTFR time and also improving the pass@1 accuracy.

3. PLANTAIN has a strong generalization ability, which enables the model to finetune on coding datasets but effectively transfer the planning ability to out-of-domain tasks like mathematical reasoning.

Weaknesses:

1. It is reported that PLANTAIN framework lowers the TTFR time. However, the authors didn’t compare the total time usage on the datasets. The contribution of the PLANTAIN framework will be minor if the total time usage is largely increased even if TTFR is reduced by 60%.

2. The authors state that PLANTAIN’s plan is evaluated by a human judge or an LLM autorater. However, the human evaluation component, which might exhibit features different from those of an LLM autorater, has not been discussed in this paper.

---

> ### Author Rebuttal · Authors · 2026-03-28
>
> ### General response to all revewers
>
> We thank the reviewers for the thoughtful feedback and are encouraged that reviewers found the paper clearly written and easy to follow (vSiF, 5Mru), also technically novel for introducing a learned plan-first post-training recipe rather than a prompting-based approach (5Mru, CFTU, K9Wz), and highlighted our strong empirical results showing substantial TTFR reduction and improved downstream performance (vSiF, 5Mru, CFTU, K9Wz). Reviewers also highlighted PLANTAIN’s ability to transfer beyond the training domain to other reasoning benchmarks including long-context QA and text-to-SQL (vSiF, K9Wz).
>
> ###  [W1] Does reducing TTFR translate to reduction in total wall-clock time?
>
> While explicit wall-clock time reporting would be useful, for a fixed model/hardware configuration, inference time is largely driven by autoregressive decoding, so total generation time scales roughly linearly with token count. Thus, **T/P already serves as a useful proxy for wall-clock cost**. Here are token/problem statistics, which we will include in the next version of the paper:
>
>
> | Model       | Method                | MATH500 | MBPP   | Text-to-SQL | QuaLITY |
> |------------|----------------------|--------:|-------:|------------:|--------:|
> | Qwen3-4B   | TA (8192)            | 3968.2  | 2816.8 | 3052.1      | 2870.2  |
> |            | Plan-Answer          | **2348.0**  | 2328.6 | **1902.4**      | 1850.2  |
> |            | R&R (Plan, 8192)     | 2496.2  | **2258.4** | 2068.3      | **1814.5**  |
> |            | Best-of-N (5, 8192)  | 2834.6  | 3022.8 | 2312.4      | 2684.2  |
> | Qwen3-8B   | TA (8192)            | 4298.9  | 2986.8 | 3214.6      | 3025.6  |
> |            | Plan-Answer          | **2654.6**  | **2489.4** | **1806.2**      | 1955.8  |
> |            | R&R (Plan, 8192)     | 2864.1  | 2492.6 | 2268.9      | **1908.2**  |
> |            | Best-of-N (5, 8192)  | 3122.9  | 3115.4 | 2428.7      | 2698.6  |
>
> Relative to TA (8192), the best plan-based settings reduce token usage by roughly 35–45% in several cases, which would imply a similar reduction in decoding wall-clock time, aside from external judge latency or other system overhead. **The tokens-per-problem (T/P) statistics show that PLANTAIN’s plan-based methods are often substantially more token-efficient than Think-Answer**
>
> When also considering pass@1 accuracy, we find that the plan-based methods often achieve higher accuracy with fewer total generated tokens than the Think-Answer baselines. In particular, for both 4B and 8B models, Plan-Answer and R&R (Plan) usually use far fewer tokens per problem than Think-Anwer (TA) (8192) while maintaining or improving accuracy. We hypothesize that this is because formulating and then conditioning on a high-level plan enables reasoning models to use thinking tokens more efficiently in a goal-directed manner. We will include this analysis in the final version of the paper.
>
> ### [W2] Effect of human judge versus LLM judge
>
> Thank you for raising this point! We agree that human judges and LLM autoraters can differ in important ways, and we will clarify these tradeoffs in the paper. Human judges may place greater weight on factors such as clarity, usefulness, trust, cognitive load, and subjective preference. In contrast, an LLM autorater may be more sensitive to prompt phrasing or stylistic cues. Humans may also be more forgiving of incomplete but interpretable plans, or conversely may reject plans that are technically plausible but confusing or hard to follow. We will revise the paper to make this distinction explicit and to note that human evaluation is an important future direction for validating the benefits of PLANTAIN on user-experience.
>
> ### [Q1/Q2] Effect of running autorater judge in real-time deployment latency
>
> Thank you for raising this point! We will clarify that the reported TTFR is defined as the time until the initial plan is shown to the user, and therefore does not include any later autorater call. If an external LLM judge is used, its latency is incurred after the first plan is already visible, so it affects end-to-end time to the final answer rather than time to first response. Since **the plans are short and judge outputs are just a few tokens**, we expect this overhead to be modest relative to full reasoning generation, although the exact latency depends on the serving and network setup. We will make this distinction explicit in the paper.

---

> > ### Author Rebuttal · Reviewer_K9Wz · 2026-04-03
> >
> > Thank you for your detailed response, which has addressed some of my concerns. Based on this, I will maintain my current rating.

---

> > > ### Author Response · Authors · 2026-04-07
> > >
> > > We thank the reviewer for their thoughtful feedback and will incorporate these suggestions into the final camera-ready version of the paper.

---

### Official Review · Reviewer_CFTU · 2026-03-12

**Soundness:** 2
**Presentation:** 2
**Significance:** 2
**Originality:** 3
**Overall Recommendation:** 3
**Confidence:** 4

**Summary:**

The paper proposes the PLANTAIN framework, which improves the conventional think-then-answer reasoning paradigm by introducing a plan-first interleaved reasoning structure. To enable this capability, the authors synthesize interleaved reasoning trajectories using LLMs and perform post-training with a combination of SFT and PPO, allowing the model to learn this structured reasoning pattern. Experimental results demonstrate that the proposed approach reduces Time-To-First-Response (TTFR) while achieving improved task accuracy.

**Compliance With Llm Reviewing Policy:**

Affirmed.

**Final Justification:**

I thank the authors for the detailed rebuttal and the additional comparison with Interleaved Reasoning. The authors also acknowledge the importance of disentangling the contributions of the key components of PLANTAIN, but this issue has still not been fully resolved in the current version, particularly due to the lack of sufficiently systematic ablations. Therefore, although I find the work promising, I will maintain my current score of 3.

**Key Questions For Authors:**

1. Does the proposed method remain effective on larger-scale models or models with different architectures?

2. Could the evaluation be extended to more challenging reasoning benchmarks such as AIME or GPQA?

3. Could the authors compare PLANTAIN with stronger or more recent baselines to better demonstrate the effectiveness of the proposed approach?

4. Could the authors provide ablation studies or further analysis to clarify the contribution of each component of the method? The framework includes multiple components, such as the plan-first reasoning structure, the interleaved reasoning format, SFT+PPO training, and plan-level inference control.

5. How sensitive are the results to the choice of the LLM judge used for evaluating candidate plans? Would the performance remain consistent if different judge models were used?

**Limitations:**

yes

**Strengths And Weaknesses:**

Strengths
1. The paper proposes the PLANTAIN framework, which introduces a plan–answer interleaving mechanism that decomposes the reasoning process into two stages: explicit planning followed by subsequent reasoning. This formulation makes the reasoning process more structured, enables potential early intervention during LLM reasoning, and reduces the time-to-first-response (TTFR).

2. The paper synthesizes interleaved reasoning traces using an LLM and applies post-training with SFT and PPO to enable the model to learn a plan-first reasoning pattern, providing a potential approach for training models to follow structured reasoning processes.

Weaknesses
1. The experiments are primarily conducted on relatively small to medium-sized models, such as Qwen3-4B and Qwen3-8B. For larger reasoning models (e.g., 30B or above), it remains unclear whether an explicit plan-first reasoning structure is still necessary. Therefore, the effectiveness of the proposed method on larger models remains uncertain.

2. The evaluation is mainly conducted on moderately difficult benchmarks such as MATH500, while more challenging reasoning benchmarks (e.g., AIME or GPQA) are not included. As a result, it remains unclear whether the proposed method can effectively handle more complex multi-step reasoning tasks.

3. The baselines in Tables 1–3 mainly consist of relatively simple methods such as Think-Answer and No Thinking, while stronger baselines are missing. This makes it difficult to fully assess the effectiveness of the proposed approach.

4. The proposed method includes several components, such as the plan-first reasoning structure, the interleaved reasoning format, SFT + PPO post-training, and plan-level inference control. However, the paper does not provide systematic ablation experiments to analyze the contribution of each component. As a result, it remains unclear which part of the method primarily leads to the observed performance improvements.

5. The proposed inference strategies, such as Best-of-N Plans and Rewind-and-Repeat, rely on an LLM judge to select or reject candidate plans. However, the paper does not analyze how the choice of judge model or potential evaluation bias affects the results. Therefore, the stability and robustness of the method under different judge settings remain unclear.

---

> ### Author Rebuttal · Authors · 2026-03-28
>
> ### General response to all reviewers
>
> We thank the reviewers for the thoughtful feedback and are encouraged that reviewers found the paper clearly written and easy to follow (vSiF, 5Mru), also technically novel for introducing a learned plan-first post-training recipe rather than a prompting-based approach (5Mru, CFTU, K9Wz), and highlighted our strong empirical results showing substantial TTFR reduction and improved downstream performance (vSiF, 5Mru, CFTU, K9Wz). Reviewers also highlighted PLANTAIN’s ability to transfer beyond the training domain to other reasoning benchmarks including long-context QA and text-to-SQL (vSiF, K9Wz).
>
> ### [W1] Evaluating PLANTAIN on larger reasoning models
>
> Thank you for this point! We agree that evaluating PLANTAIN at larger scales would be valuable for understanding how the approach behaves for stronger reasoning models. However, such an evaluation would require RL post-training at 30B+ scale, which is prohibitively expensive under our compute budget and is still relatively uncommon. We therefore focus on 4B and 8B models as a controlled setting to test the core scientific question. These model sizes are also among the most widely used and deployed open models, making them a relevant and representative setting.
>
> Importantly, our claim is not that larger models are unable to reason without explicit plans. Rather, surfaced plans provide a useful interface for control and intervention, and that PLANTAIN can be learned effectively via post-training open-model scales. We will clarify this scope more explicitly in the paper and note larger-scale evaluation as an important direction for future work.
>
> ### [W2] Evaluating on more difficult reasoning benchmarks
>
> Thank you for this suggestion! We agree that including harder reasoning benchmarks such as AIME or GPQA would further strengthen the evaluation. Our current benchmark suite already establishes that PLANTAIN transfers beyond coding to text-to-SQL and long-context QA, demonstrating cross-domain generality of the plan-first training recipe.
>
> We expect the benefits of PLANTAIN to be even more pronounced on harder multi-step reasoning tasks. As problems become more complex, successful solving depends increasingly on early decomposition and on avoiding suboptimal reasoning paths that are expensive to correct later. A plan-first interface is particularly well suited to this regime, since it surfaces the intended solution structure before the model commits to detailed reasoning. We will consider extending the evaluation with harder reasoning benchmarks in the camera-ready version / final release.
>
> ### [W3] More state-of-the-art baselines
>
> Thank you for this suggestion! The baselines were chosen as controlled comparisons to test plan-first reasoning with and without PLANTAIN. No Thinking and Think-Answer are intended as reference points for measuring the effect of adding planning structure. If the reviewer has specific additional baselines in mind, we would greatly appreciate those suggestions and will consider adding them in the next version.
>
> ### [W4] Ablations on each component of PLANTAIN
>
> Thank you for this suggestion! In the current version, we already ablate the plan-level inference-control by comparing direct Plan-Answer, Best-of-N planning, Rewind-and-Repeat at the plan level, and Rewind-and-Repeat at the final-answer level. While other ablations are possible, we believe that the existing ablations already demonstrate the effectiveness of plan-level inference control.
>
> We hypothesize that SFT teaches the model the desired plan-first / interleaved response format and strengthens instruction-following for surfacing plans, while PPO further improves final-task performance conditioned on the plan by optimizing verifiable rewards. This is also consistent with recent reasoning-model pipelines that combine cold-start SFT with RL, such as Qwen3, and with reports that RL can provide additional gains on verifiable reasoning tasks beyond supervised training alone. That said, we do not explicitly decouple SFT-only vs. SFT+PPO in the current paper, and we agree that this should be isolated in future ablations.
>
> ### [W5] Robustness study with different LLM judges
>
> Thank you for the suggestion! We agree that the choice of LLM autorater can affect the downstream task performance. We will clarify in the paper that the current experiments are conducted under a fixed user simulator. A more complete robustness study over the choice of LLM autorater and prompt would be valuable, as would experiments with human feedback or lightweight task-specific verifiers. However, we do not believe this weakens the main contribution of the paper, which is the plan-first post-training recipe and inference-time generation mechanisms. Since all methods are evaluated with the same model, the comparison is controlled, and the reported gains reflect the relative effectiveness of the inference strategies.

---

> > ### Author Rebuttal · Reviewer_CFTU · 2026-04-02
> >
> > Thank you for the detailed rebuttal. Several of my earlier concerns have been reasonably addressed, and some of the remaining ones may be further clarified or supported with additional experiments in a future version. I also find this work interesting.
> >
> > That said, among the remaining concerns, I believe the following are the most important points to prioritize for strengthening the paper, ideally with additional experimental validation:
> >
> >
> > [W3] More baselines
> >
> > In particular, it would be valuable to include comparisons with concurrent PPO-based interleaved reasoning approaches[1] as well as training-free methods such as StreamingThinker[2], to better position the contribution.
> >
> > [W4] Ablations on each component of PLANTAIN
> >
> > While the current experiments provide analysis on inference-time strategies, it remains unclear how much each component, such as the plan-first structure, the interleaved format, and SFT versus PPO training, contributes to the overall performance gains. More systematic ablations would significantly strengthen the paper by isolating the key factors driving the improvements.
> >
> > References
> >
> > [1] Roy Xie, et al. Interleaved reasoning for large language models via reinforcement learning. arXiv:2505.19640, 2025.
> >
> > [2] Junlong Tong, et al. StreamingThinker: Large Language Models Can Think While Reading. arXiv:2510.17238, 2025.

---

> > > ### Author Response · Authors · 2026-04-07
> > >
> > > We thank the reviewer for suggesting comparisons to Interleaved Reasoning via RL (Xie et al., 2025) and StreamingThinker (Tong et al., 2025). We agree these are relevant directions as both explore alternatives to the standard think-then-answer paradigm.
> > >
> > > We include a direct comparison to Interleaved Reasoning via RL, as it operates in a similar setting and targets improved reasoning efficiency through interleaving. To ensure an apples-to-apples comparison, we re-ran this baseline under the same backbone, task suite, and decoding budgets used in our paper, initializing all methods from the same Qwen3-8B checkpoint. Importantly, Interleaved Reasoning via RL is an RL-only post-training method: it does not include a supervised fine-tuning stage on interleaved traces, but instead relies on reinforcement learning to induce interleaved reasoning behavior from a prompted base model.
> > >
> > > We do not include StreamingThinker in our comparison because it is designed for a fundamentally different setting: streaming-input reasoning, where the model receives the input incrementally and overlaps reading with thinking. Its primary contributions are specifically aimed at reducing latency when the input itself is still arriving. In contrast, all four of our benchmarks are static-input tasks, where the full problem is available upfront and latency is dominated by reasoning rather than input ingestion. Adapting StreamingThinker to this setting removes its core advantage and results in a model that is no longer aligned with its intended use case.
> > >
> > > Across all four tasks, we find that PLANTAIN outperforms Interleaved Reasoning. Interleaved Reasoning improves efficiency by training the model to interleave reasoning and answer generation within a single trajectory, but it still commits to that trajectory from the start and does not support revising the overall approach before continuing. PLANTAIN, by contrast, produces an execution plan first, which can be verified, rejected, or revised before full reasoning begins. Combined with Rewind-and-Repeat (R&R), this lets PLANTAIN explore multiple candidate plans and spend additional compute on selecting a better plan before committing to full reasoning. Empirically, this leads to similar or better T/P together with higher accuracy, especially on Text-to-SQL and QuaLITY, where early high-level decisions are particularly important.
> > >
> > > | Method                             |  MATH500 |     MBPP | Text-to-SQL |  QuaLITY |
> > > | ---------------------------------- | -------: | -------: | ----------: | -------: |
> > > | Think-Answer (8192)                |     88.2 |     56.4 |        31.0 |     78.4 |
> > > | Interleaved Reasoning (Xie et al.) |     87.8 |     57.1 |        31.8 |     77.6 |
> > > | Plan-Answer (ours)                 |     85.2 |     56.8 |        28.0 |     76.4 |
> > > | Best-of-N (5, 8192) (ours)         |     89.4 |     58.7 |        32.5 | **78.8** |
> > > | R&R (Plan, 8192) (ours)            | **89.6** | **59.1** |    **34.3** |     78.4 |
> > >
> > > | Method                                             |    MATH500 |       MBPP | Text-to-SQL |    QuaLITY |
> > > | -------------------------------------------------- | ---------: | ---------: | ----------: | ---------: |
> > > | Think-Answer (8192)                                |     4298.9 |     2986.8 |      3214.6 |     3025.6 |
> > > | Interleaved Reasoning (Xie et al.) |     3010.0 |     2460.0 |      2380.0 |     2290.0 |
> > > | Plan-Answer (ours)                                 | **2654.6** | **2489.4** |  **1806.2** |     1955.8 |
> > > | Best-of-N (5, 8192) (ours)                         |     3122.9 |     3115.4 |      2428.7 |     2698.6 |
> > > | R&R (Plan, 8192) (ours)                            |     2864.1 |     2492.6 |      2268.9 | **1908.2** |

---

### Official Review · Reviewer_5Mru · 2026-03-13

**Soundness:** 4
**Presentation:** 3
**Significance:** 3
**Originality:** 2
**Overall Recommendation:** 5
**Confidence:** 3

**Summary:**

The authors propose a training recipe that encourages the model to produce an explicit plan as the first intermediate response, thereby reducing the perceived latency, as defined by the number of tokens until the first intermediate response:
1. generation of synthetic interleaved reasoning traces
2. supervised fine-tuning to distill the interleaved format
3. RL post-training using verifiable rewards
This recipe is complemented by two inference-time strategies:
1. Best-of-N plans
2. Rewind-and-repeat
Experiments across Qwen models and diverse datasets (MATH500, MBPP, Text-to-SQL, and QuaLITY) demonstrate the effectiveness of their approach across math, coding, and question-answering.

**Compliance With Llm Reviewing Policy:**

Affirmed.

**Final Justification:**

Missing T/P results added during rebuttal, adding to the empirical soundness.

**Key Questions For Authors:**

Q1: Could you point me to the tokens-per-problem results? (I see it mentioned in Section 4, but couldn't find it in any of the results tables.) This would further strengthen the claim (S2 from above) that Plantain improves overall efficiency, not just the latency as defined by the time to first response.

**Limitations:**

Yes

**Strengths And Weaknesses:**

S1: TTFR is consistently reduced across all experiments. For example, in Table 1, the TTFR for Qwen3-4B with Plantain drops from 1419 to 516 on average across datasets, while accuracy (P@1) increases from 56.9 to 58.8. The accuracy is also considerably higher than the no thinking baseline of 52.4.

S2: In addition to reducing the TTFR, overall token usage appears to be reduced as well from Table 3. In particular, the accuracy of R&R with 4096 token budget is higher than the accuracy of Think-Answer with 8192 token budget.

S3: The presentation and writing in the main paper is clear. (The abstract could better emphasize the fact that method is a training recipe. My initial impression was that it was purely prompting based.)

S4: The paper proposes a novel training recipe to improve planning capabilities via training. All plan-then-execute methods that I know of are prompting-based.

W1: Without user studies measuring perceived latency, I'm not fully convinced that Plaintain reduces "perceived latency," which was the original motivation in the abstract: "By providing useful information to the user earlier, IR reduces perceived latency, the time a user waits for an initial output, without compromising the quality of the final response." I don't see this is an issue with the contributions/experiments though, just something that could be improved about the initial framing.

---

> ### Author Rebuttal · Authors · 2026-03-28
>
> ### General response to all reviewers
>
> We thank the reviewers for the thoughtful feedback and are encouraged that reviewers found the paper clearly written and easy to follow (vSiF, 5Mru), also technically novel for introducing a learned plan-first post-training recipe rather than a prompting-based approach (5Mru, CFTU, K9Wz), and highlighted our strong empirical results showing substantial TTFR reduction and improved downstream performance (vSiF, 5Mru, CFTU, K9Wz). Reviewers also highlighted PLANTAIN’s ability to transfer beyond the training domain to other reasoning benchmarks including long-context QA and text-to-SQL (vSiF, K9Wz).
>
> ###  [S3] PLANTAIN is a post-training recipe for plan-first generation
>
> Thank you for pointing this out! We agree that the abstract should more clearly present **PLANTAIN as a training recipe / post-training method, rather than a purely prompting-based approach**. We will revise it accordingly in the camera-ready version. We will clarify that PLANTAIN post-trains the model to externalize a plan before continuing the reasoning process. The model first outputs an explicit plan, and then carries out subsequent reasoning conditioned on that surfaced plan. We will update the abstract to make this training-based framing explicit, so readers immediately understand that the contribution is a learned inference-time interface enabled by post-training and not just a prompting strategy.
>
> ### [W1] Framing PLANTAIN about reducing perceived latency
>
> Thank you for this helpful framing suggestion! We agree that, without a user study, we should avoid making a strong empirical claim about perceived latency. Our empirical results support a narrower claim: PLANTAIN reduces TTFR, i.e., the delay until the first useful surfaced response, while maintaining or improving downstream task performance. We will revise the framing accordingly in the camera-ready version to distinguish between measured TTFR reduction and perceived latency as a motivating interpretation. In particular, we will present improved user experience as the motivation for surfacing plans earlier. We agree with the reviewer that this does not weaken the main contribution, which is the plan-first post-training recipe and its empirical improvement that improves the latency–accuracy tradeoff.
>
> ### [Q1] Tokens per problem (T/P) analysis
>
> | Model       | Method                | MATH500 | MBPP   | Text-to-SQL | QuaLITY |
> |------------|----------------------|--------:|-------:|------------:|--------:|
> | Qwen3-4B   | TA (8192)            | 3968.2  | 2816.8 | 3052.1      | 2870.2  |
> |            | Plan-Answer          | **2348.0**  | 2328.6 | **1902.4**      | 1850.2  |
> |            | R&R (Plan, 8192)     | 2496.2  | **2258.4** | 2068.3      | **1814.5**  |
> |            | Best-of-N (5, 8192)  | 2834.6  | 3022.8 | 2312.4      | 2684.2  |
> | Qwen3-8B   | TA (8192)            | 4298.9  | 2986.8 | 3214.6      | 3025.6  |
> |            | Plan-Answer          | **2654.6**  | **2489.4** | **1806.2**      | 1955.8  |
> |            | R&R (Plan, 8192)     | 2864.1  | 2492.6 | 2268.9      | **1908.2**  |
> |            | Best-of-N (5, 8192)  | 3122.9  | 3115.4 | 2428.7      | 2698.6  |
>
> **The tokens-per-problem (T/P) results show that PLANTAIN’s plan-based methods are often substantially more token-efficient than Think-Answer**. In particular, for both 4B and 8B models, Plan-Answer and R&R (Plan) usually use far fewer tokens per problem than Think-Anwer (TA) (8192) while maintaining or improving accuracy. On QuaLITY and Text-to-SQL especially, the gap is large: e.g., for Qwen3-8B, Think-Answer (8192) uses 3025.6 / 3214.6 T/P, versus 1908.2 / 1806.2 for the best plan-based settings. We hypothesize that this is because formulating and then conditioning on a high-level plan enables reasoning models to use thinking tokens more efficiently in a goal-directed manner. We will include this analysis in the final version of the paper.

---

> > ### Author Rebuttal · Reviewer_5Mru · 2026-04-04
> >
> > Thank you for including the tokens per problem (T/P) analysis, which is equally as convincing as the TTFR results in the original submission and adds to the soundness of the experiments. I will increase my to solid accept.

---

> > > ### Author Response · Authors · 2026-04-07
> > >
> > > We thank the reviewer for their thoughtful feedback and are glad that the added T/P analysis was helpful in addressing their concern. We appreciate their positive assessment and will incorporate these suggestions into the final camera-ready version of the paper.

---

### Official Review · Reviewer_vSiF · 2026-03-18

**Soundness:** 4
**Presentation:** 3
**Significance:** 3
**Originality:** 3
**Overall Recommendation:** 5
**Confidence:** 4

**Summary:**

This paper proposes an alternate strategy to think-then-answer approach to reasoning, called PLANTAIN.

Instead of generating full monolithic reasoning (thinking tokens), PLANTAIN (this work) generates a short plan first, which can be accepted or rejected by the user. If accepted, it will then generate the thinking tokens and then the final answer. If rejected, it will generate a new plan, and the process repeats until the user accepts a plan. This is PLANTAIN's Rewind-and-Repeat variant. Alternatively, the Best-of-N-Plans variant generates N plans, and then the user selects the one.

The authors develop a post-training pipeline to make LLMs capable of following PLAINTAIN's plan-first approach. It involves SFT with different kinds of curated data from LLMs and RL with verifiable rewards on format, correctness, helpfulness, and task outcome (unit tests). Both done on BIGCODEBENCH.

Authors post-train QWEN3 4B and 8B and demonstrate (using QWEN2.5 as simulated user/judge) that not only reduces latency (for the first response), but also improves model performance across several benchmarks. I.e., Plan-Answer > Think-Answer. Surprisingly, even though post-training is done on code generation, the latency and performance gains also generalize to other domains.

**Compliance With Llm Reviewing Policy:**

Affirmed.

**Key Questions For Authors:**

No questions; see review.

**Limitations:**

yes

**Strengths And Weaknesses:**

# Strengths

* Consistent improvements in performance and first response latency across several benchmarks.
* Good generalization to even domains the approach was not trained on.
* Good comparative results on which inference-time strategy is better.
* The paper is clearly written and easy to follow.

# Weaknesses

This is a strong paper with good results. The points below are not necessarily weaknesses, but rather suggestions for further improvement.

- One of the motivations of PLAINTAIN is to have a better user experience. Intuitively, it makes sense that users could prefer plan-first approach. But it would be good to have a user study to evaluate and compare the perceived experience with the baseline. Do users actually prefer plans provided by PLAINTAIN over the direct thinking tokens?

- It would be nice to extend the evaluation to agentic tasks (e.g., tool use, multi-step interactions), where long-range planning is quite important. It'd be interesting to see if PLAINTAIN's plans are helpful for such tasks as well.

- Besides TTFR, it would be good to also show the average total tokens (even in the Appendix), indicating whether PLAINTAIN is more efficient or not (I do see Table 3, but it's unclear what average tokens are for Table 1).

---

> ### Author Rebuttal · Authors · 2026-03-28
>
> ### General response to all reviewers
>
> We thank the reviewers for the thoughtful feedback and are encouraged that reviewers found the paper clearly written and easy to follow (vSiF, 5Mru), also technically novel for introducing a learned plan-first post-training recipe rather than a prompting-based approach (5Mru, CFTU, K9Wz), and highlighted our strong empirical results showing substantial TTFR reduction and improved downstream performance (vSiF, 5Mru, CFTU, K9Wz). Reviewers also highlighted PLANTAIN’s ability to transfer beyond the training domain to other reasoning benchmarks including long-context QA and text-to-SQL (vSiF, K9Wz).
>
> ### [W1] User study for evaluating effect of PLANTAIN on user experience
>
> Thank you for the suggestion! We agree that a user study would be valuable, especially since a key motivation of PLANTAIN is to improve user experience by surfacing a plan early and enabling intervention before the model commits to a potentially suboptimal reasoning path. We did not include a human preference study in the current submission, but we will add this as an important direction for future work and will strongly consider including such an evaluation in the camera-ready version.
>
> ### [W2] Evaluation on agentic benchmarks
>
> Thank you for this suggestion! We agree that agentic tasks are a particularly compelling setting for PLANTAIN, since surfacing a plan is likely even more valuable when later actions are costly or hard to undo. In tool-use and multi-step interaction settings, early inspection and correction of the plan can be especially important. Our current experiments focus on a controlled evaluation across several domains, including coding, math, and structured queries, to isolate the core effect of the method. We agree that extending the evaluation to agentic settings would strengthen the paper, and we will consider this for the camera-ready version, potentially using benchmarks such as AgentBench [1] or OSWorld [2].
>
> ### [W3] Tokens per problem (T/P) analysis
>
> | Model       | Method                          | MATH500 | MBPP   | Text-to-SQL | QuaLITY |
> |------------|----------------------------------|--------:|-------:|------------:|--------:|
> | Qwen3-4B   | Think-Answer (8192)             | 3968.2  | 2816.8 | 3052.1      | 2870.2  |
> |            | Plan-Answer                     | **2348.0** | 2328.6 | **1902.4** | 1850.2  |
> |            | Rewind-and-Repeat (Plan, 8192)  | 2496.2  | **2258.4** | 2068.3  | **1814.5** |
> |            | Best-of-N (5, 8192)             | 2834.6  | 3022.8 | 2312.4      | 2684.2  |
> | Qwen3-8B   | Think-Answer (8192)             | 4298.9  | 2986.8 | 3214.6      | 3025.6  |
> |            | Plan-Answer                     | **2654.6** | **2489.4** | **1806.2** | 1955.8  |
> |            | Rewind-and-Repeat (Plan, 8192)  | 2864.1  | 2492.6 | 2268.9      | **1908.2** |
> |            | Best-of-N (5, 8192)             | 3122.9  | 3115.4 | 2428.7      | 2698.6  |
>
> **The tokens-per-problem (T/P) results show that PLANTAIN’s plan-based methods are often substantially more token-efficient than Think-Answer**. In particular, for both 4B and 8B models, Plan-Answer and R&R (Plan) usually use far fewer tokens per problem than Think-Anwer (TA) (8192) while maintaining or improving accuracy. On QuaLITY and Text-to-SQL especially, the gap is large: e.g., for Qwen3-8B, Think-Answer (8192) uses 3025.6 / 3214.6 T/P, versus 1908.2 / 1806.2 for the best plan-based settings. We hypothesize that this is because formulating and then conditioning on a high-level plan enables reasoning models to use thinking tokens more efficiently in a goal-directed manner. We will include this analysis in the final version of the paper.
>
> [1] Li et al., "LiveAgentBench: Comprehensive Benchmarking of Agentic Systems Across 104 Real-World Challenges," 2026.
>
> [2] Xie et al., "OSWorld: Benchmarking Multimodal Agents for Open-Ended Tasks in Real Computer Environments", 2025.

---

### Decision · Program_Chairs · 2026-04-30

**Decision:**

Accept (spotlight)

**Comment:**

This paper introduces PLANTAIN with interleaved reasoning to replace the standard think-then-answer reasoning process in current models. PLANTAIN trains models to output an explicit, step-by-step plan before completing the full reasoning process. Two inference-time strategies, Best-of-N-Plans and Rewind-and-Repeat, were also proposed. Most reviewer gave positive scores on the submission. The main concerns are as follows.

1. The evaluation on user experience could be better for the paper.
2. It is better to conduct more comparisons with concurrent PPO-based interleaved reasoning approaches and ablations studies.